# WS-GRPO: Weakly-Supervised Group-Relative Policy Optimization for Rollout-Efficient Reasoning

Gagan Mundada [* 1]  Zihan Huang [* 1]  Rohan Surana [* 1]  Sheldon Yu [1]  Jennifer Yuntong Zhang [2]  Xintong Li [1]  Tong Yu [3]  Lina Yao [4 5]  Jingbo Shang [1]  Julian McAuley [1]  Junda Wu [† 1]

## Abstract

Group Relative Policy Optimization (GRPO) is effective for training language models on complex reasoning. However, since the objective is defined relative to a group of sampled trajectories, extended deliberation can create more chances to realize relative gains, leading to inefficient reasoning and overthinking, and complicating the trade-off between correctness and rollout efficiency. Controlling this behavior is difficult in practice, considering (i) Length penalties are hard to calibrate because longer rollouts may reflect harder problems that require longer reasoning, penalizing tokens risks truncating useful reasoning along with redundant continuation; and (ii) supervision that directly indicates when to continue or stop is typically unavailable beyond final answer correctness. We propose Weakly Supervised GRPO (WS-GRPO), which improves rollout efficiency by converting terminal rewards into correctness-aware guidance over partial trajectories. Unlike global length penalties that are hard to calibrate, WS-GRPO trains a preference model from outcome-only correctness to produce prefix-level signals that indicate when additional continuation is beneficial. Thus, WS-GRPO supplies outcome-derived continue/stop guidance, reducing redundant deliberation while maintaining accuracy. We provide theoretical results and empirically show on reasoning benchmarks that WS-GRPO substantially reduces rollout length while remaining competitive with GRPO baselines.

*Equal contribution. †Corresponding author. [1]University of California, San Diego [2]University of Toronto [3]Adobe Research [4]The University of New South Wales [5]CSIRO's Data61. Correspondence to: Junda Wu <juw069@ucsd.edu>.

*Proceedings of the 43rd International Conference on Machine Learning*, Seoul, South Korea. PMLR 306, 2026. Copyright 2026 by the author(s).

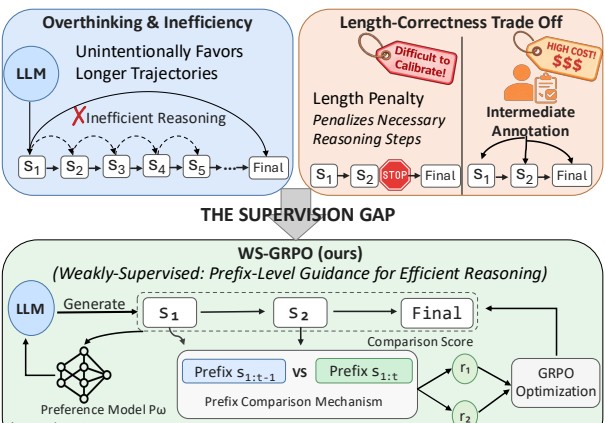

*Figure 1.* **WS-GRPO for efficient reasoning.** GRPO's group-relative objective can favor longer trajectories, while simple length penalties are hard to calibrate and human supervision is expensive (top). WS-GRPO uses a preference model trained on correct vs. incorrect outcomes to compare consecutive prefixes, generating correctness-aware guidance that reduces redundant continuation while preserving necessary reasoning (bottom). Some icons were generated by a generative AI tool (ChatGPT) and are for illustrative purposes only.

## 1. Introduction

Large language models (LLMs) have advanced in complex reasoning (Wei et al., 2022; Wang et al., 2026; Yu et al., 2025; Wu et al., 2026; 2024; 2025a;c; Xia et al., 2025b), and GRPO further improves their training by replacing value-function learning with group-normalized advantages, enhancing stability, sample efficiency, and memory usage (Shao et al., 2024). Despite these benefits, GRPO can unintentionally encourage extended deliberation(Snell et al., 2025; Miao et al., 2024), and models may generate unnecessarily long trajectories that increase computation and lead to overthinking (Snell et al., 2025; Miao et al., 2024). Achieving a good trade-off between correctness and rollout efficiency is difficult in practice: simple length penalties are often inadequate because trajectory length correlates with problem difficulty and solution validity (Dubois et al., 2024; Saito et al., 2023), while fine-grained supervision that could identify low-utility continuation is typically unavailable beyond final-answer correctness (Lightman et al., 2024;

Uesato et al., 2022).

To bridge such trade-offs, a common approach is to impose token-level length regularization (Kikuchi et al., 2016; Murray & Chiang, 2018). However, such global penalties are often difficult to calibrate and can be unreliable when length correlates with instance difficulty and solution validity (Dubois et al., 2024), as penalizing tokens may suppress necessary deliberation on hard problems as well as redundant continuation on easy ones (Snell et al., 2025). Addressing this issue more directly would require supervision that can distinguish promising continuation from low-utility continuation during generation. However, collecting such fine-grained intermediate annotations is costly and difficult to scale beyond outcome-level signals such as final-answer correctness (Lightman et al., 2024; Uesato et al., 2022).

Weakly supervised learning provides a principled way to use coarse outcome signals when fine-grained intermediate supervision is unavailable (Zhou, 2018; Zhang et al., 2024; Wu et al., 2022; 2023). For GRPO, however, a trajectory-level correctness label does not indicate when additional generation improves the probability of a correct solution, and many different continuations can lead to the same final outcome (Zelikman et al., 2024; Wang et al., 2023). As a result, redundant continuation and necessary reasoning can be equally consistent with correctness, so outcome-based intermediate incentives are poorly identified and often noisy. This issue is amplified under policy updates, where distribution shift can further destabilize intermediate proxies derived from outcomes. These observations motivate structuring weak supervision around partial trajectories, so that supervision targets the value of continuing from a given prefix rather than treating length or intermediate tokens as directly supervised.

In this work, we introduce Weakly Supervised GRPO (WS-GRPO), which improves rollout efficiency using weak supervision that provides correctness aware, prefix level guidance. WS-GRPO avoids global length penalties and instead estimates the marginal value of continuing a trajectory by comparing successive partial trajectories using a preference model trained from outcome correctness. This yields dense training signals that downweight continuation when additional generation provides limited utility, while preserving continuation that is necessary to reach a correct solution. WS-GRPO proceeds in two stages: (i) it learns a preference model that discriminates between correct and incorrect reasoning trajectories from outcome labels; and (ii) it uses this model to compare consecutive prefixes within each rollout and converts the resulting preference margins into prefix level pseudo rewards, which are combined with terminal correctness in the GRPO objective to encourage earlier formation of high quality prefixes and reduce unnecessary continuation. Our contributions are as follows:

- We derive correctness-aware prefix weak supervision from terminal rewards by converting preference margins between consecutive prefixes into dense signals.

- We propose WS-GRPO, a two-stage method that learns a trajectory-level preference model from correct/incorrect rollouts and integrates prefix-level pseudo-rewards with terminal correctness in GRPO.

- We provide theoretical results on the consistency of the weakly supervised preference model, a preference-error controlled robustness bound, and a high-probability generalization bound.

- We evaluate WS-GRPO on reasoning benchmarks, demonstrating substantially reduces on rollout length while remaining competitive with strong baselines, yielding more concise and reliable solutions.

## 2. Preliminaries

### 2.1. Weakly-Supervised Learning

Weakly-supervised learning (WSL) studies settings where supervision is inexact or implicit, so dense instance-level labels are unavailable (Zhou, 2018; Wu et al., 2022; 2023). In our setting, supervision is typically limited to outcome correctness, which is informative about overall solution quality but does not identify when continued generation is useful versus redundant. We therefore represent weak supervision through pairwise preferences, which provide a relative signal for comparing partial outputs under the same prompt. Given $(x_A, x_B) \in \mathcal{X}^2$, we observe a weak label $y \in 0, 1$ indicating whether $x_A$ is preferred to $x_B$, and learn a preference model $P_\theta : \mathcal{X}^2 \to [0, 1]$ that estimates the posterior preference probability (Shu et al., 2020; Zhang et al., 2024; Kveton et al., 2025; Huang et al., 2025c;a):

$$P_\theta(x_A, x_B) = P(x_A \succ x_B \mid x_A, x_B; \theta). \quad (1)$$

To align the model with the weak supervision, we minimize the empirical risk under a Bradley-Terry-type negative log-likelihood (NLL) objective:

$$\mathcal{L}_{\text{pref}} = \mathbb{E}_{(x_A, x_B, y)} \bigg[ - y \log P_\theta(x_A, x_B) \\ - (1-y) \log(1 - P_\theta(x_A, x_B)) \bigg]. \quad (2)$$

By optimizing over these comparative signals, the model learns to map coarse-grained supervision to latent cardinal scores, enabling fine-grained assessment in domains like process-oriented supervision where absolute labels are often unavailable or ambiguous (Cui et al., 2026).

## 2.2. Group-Relative Policy Optimization (GRPO)

Given a prompt $q$, GRPO samples $G$ independent rollouts $\{\tau_i\}_{i=1}^{G}$ from policy $\pi_\theta$, where each rollout receives scalar return $R_i = R_\phi(q, \tau_i)$. The group-relative advantage is computed as:

$$\hat{A}_i = \frac{R_i - \bar{R}}{\sigma_R}, \ \bar{R} = \frac{1}{G} \sum_{i=1}^{G} R_i, \ \sigma_R = \sqrt{\frac{1}{G} \sum_{i=1}^{G} (R_i - \bar{R})^2}.$$

The GRPO objective uses PPO-style clipping with probability ratios and KL regularization:

$$
J_{\text{GRPO}}(\theta) = \mathbb{E}_{q, \{\tau_i\}} \Big[ \frac{1}{G} \sum_{i=1}^{G} \frac{1}{|\tau_i|} \sum_{t=1}^{|\tau_i|} \min\Big( \rho_{i,t}(\theta) \hat{A}_i, \\
\text{clip}(\rho_{i,t}(\theta), 1 - \epsilon, 1 + \epsilon) \hat{A}_i \Big) - \beta \, \mathcal{L}_{\text{KL}} \Big],
$$

(3)

where probability ratio and KL divergence are defined as:

$$\rho_{i,t}(\theta) = \frac{\pi_\theta(a_{i,t}|s_{i,t})}{\pi_{\text{ref}}(a_{i,t}|s_{i,t})}, \mathcal{L}_{\text{KL}} = D_{\text{KL}}(\pi_\theta \| \pi_{\text{ref}}).$$

# 3. Weakly-Supervised-Group-Relative Preference Optimization

While GRPO has shown strong performance on multi-step reasoning tasks (Shao et al., 2024; Guo et al., 2025; Wu et al., 2026; Yu et al., 2025), its group-relative objective can unintentionally favor overly long rollouts, since extended deliberation creates more chances to realize relative gains within a sampled group. Controlling this behavior is difficult because most settings provide only sparse outcome supervision (e.g., final-answer correctness) and lack signals that indicate when to continue or stop. With such terminal rewards, GRPO exhibits *credit diffusion*: the trajectory-level label cannot identify which intermediate decisions drive success, so the supervision is delayed and underdetermined. This yields noisy, high-variance updates that can reinforce incidental patterns (e.g., templates or verbosity) and makes naive length regularization hard to calibrate when length correlates with difficulty and validity. These limitations motivate converting outcome correctness into correctness-aware, prefix-level guidance that targets the marginal value of continuing from a given prefix rather than globally penalizing tokens.

To mitigate GRPO's tendency toward redundant deliberation under outcome-only supervision, we propose Weakly Supervised GRPO (WS-GRPO) and formalize the setting (Section 3.1). WS-GRPO converts binary final-answer correctness into correctness-aware, prefix-level guidance by training a preference model from trajectory-level outcomes

and using it to estimate the marginal value of continuing from a given prefix. As illustrated in Figure 2, we adopt a two-stage procedure: in **Phase I (Section 3.2)**, we learn a preference model to discriminate between successful and unsuccessful reasoning trajectories using only complete rollouts; the key insight is that a model assessing overall trajectory quality can be repurposed to evaluate incremental progress by comparing successive partial trajectories. In **Phase II (Section 3.3)**, we apply this model to consecutive prefixes ($s_{1:t-1}$ vs. $s_{1:t}$) within each rollout and convert the resulting preference margins into prefix-level pseudo-rewards, which are combined with terminal correctness in the GRPO objective to discourage low-utility continuation while preserving necessary deliberation.

## 3.1. Problem Formulation

We consider multi-step reasoning tasks where a language model generates a sequence of reasoning steps to solve a problem. Given a question $q$, the policy $\pi_\theta$ generates a trajectory $\tau = (s_1, s_2, \ldots, s_T)$ where each $s_t$ denotes a sentence-level natural-language reasoning step (a span of tokens delimited by sentence boundaries), rather than an individual token or a latent environment state. Let $\hat{a}_i$ denote the final answer extracted from trajectory $\tau_i$ and $a^*(q)$ denote the ground truth answer for question $q$.

GRPO samples $G$ trajectories $\{\tau_i\}_{i=1}^{G}$ for each question and computes group-relative advantages from a scalar trajectory return $R_i$ (Section 2.2), which in standard GRPO is typically the terminal outcome reward $R_i^{\text{final}}$ (e.g., a binary indicator of final-answer correctness). While this approach enables policy optimization without learned value functions, it faces a fundamental limitation of insufficient signal for effective credit assignment across reasoning steps due to sparse terminal rewards.

## 3.2. Phase I: Weakly-Supervised Preference Learning

We train a weakly supervised preference model that distinguishes successful from unsuccessful reasoning trajectory prefixes using only trajectory-level outcomes, as shown in Algorithm 1. This exemplifies weak supervision, where learning occurs from indirect labels rather than direct step-level annotations (Zhou, 2018; Wu et al., 2022; Wang et al., 2025c). For each question $q$, we sample $K$ reasoning trajectories $\{\tau_1, \ldots, \tau_K\}$, where each $\tau_i = (s_{i,1}, s_{i,2}, \ldots, s_{i,T_i})$ is evaluated using a terminal outcome signal (final-answer correctness). Following Equation (1), we then construct ordered preference pairs $(\tau_a, \tau_b)$ by pairing a correct trajectory $\tau^+$ with an incorrect trajectory $\tau^-$. Specifically, we assign label $y = 1$ to the ordered pair $(\tau^+, \tau^-)$ and label $y = 0$ to its reversed ordering $(\tau^-, \tau^+)$. We handle three scenarios: (i) *mixed outcomes* yield direct within-question pairs, (ii) *all-correct outcomes* pair each correct trajectory with incor-

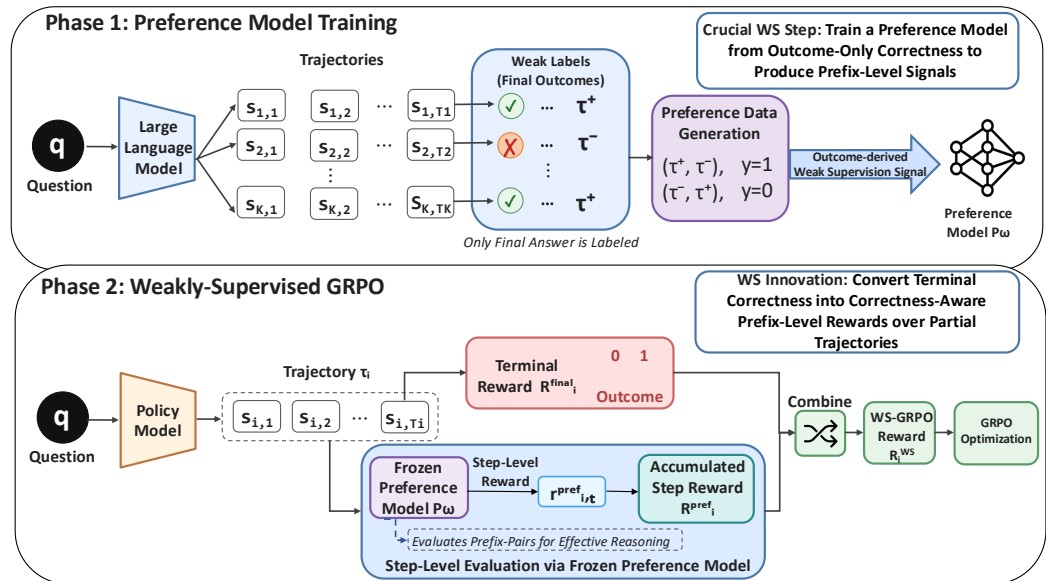

*Figure 2.* **WS-GRPO Framework Overview:** *Phase 1:* using outcome-only correctness (final-answer labels), we construct preference pairs from correct vs. incorrect rollouts and train a preference model $P_\omega$. *Phase 2:* the frozen $P_\omega$ converts terminal correctness into correctness-aware, prefix-level rewards by comparing consecutive prefixes within each rollout. These dense prefix-level rewards are combined with the terminal reward and used in the GRPO objective to refine the policy, improving rollout efficiency.

rect trajectories from other questions, and (iii) *all-incorrect outcomes* pair each incorrect trajectory with correct trajectories from other questions, where cross-question trajectories serve as negative trajectories, since they are irrelevant to the target question.

The preference model instantiates Equation (1) by converting outcome-only correctness into a scalable pairwise supervision signal that treats a correct trajectory as preferred to an incorrect one for the same question. It learns a notion of continuation utility that we later reuse to assess the marginal value of continuing from partial trajectories. The preference model encodes question $q$ jointly with both the correct trajectory $\tau^+$ and incorrect trajectory $\tau^-$, using a structured natural language template that presents both reasoning chains as options for comparison. The template explicitly asks the model to compare the quality of the two reasoning approaches in the context of the given question. This combined input is processed by an FLAN-T5 encoder, producing contextualized hidden states $\mathbf{H} \in \mathbb{R}^{L \times d}$. We extract the first token's representation $h = \mathbf{H}[0]$ as a global summary of the comparison context. A lightweight MLP classifier $P_\omega : \mathbb{R}^d \to \mathbb{R}$ then processes this representation to produce a preference score:

$$\hat{y} = P_\omega(q, \tau^+, \tau^-), \qquad (4)$$

which outputs the probability that $\tau_A$ is preferred over $\tau_B$. We train with label $\hat{y} = 1$ for the correct trajectory $\tau^+$.

We train this preference model following the Bradley-Terry objective from Equation (2) using binary cross-entropy loss. Since the training dataset contains preference pairs in both

orientations (i.e., both $(\tau^+, \tau^-)$ with label $y = 1$ and $(\tau^-, \tau^+)$ with label $y = 0$), the training objective becomes:

$$\mathcal{L}_{\text{pref}} = \mathbb{E}\Big[\text{BCE}(P_\omega(q, \tau^+, \tau^-), 1) \\ + \text{BCE}(P_\omega(q, \tau^-, \tau^+), 0)\Big], \qquad (5)$$

where each pair is encoded in both orderings to ensure symmetric preference learning regardless of input position.

This training procedure produces a preference model that captures reasoning quality patterns from outcome-level supervision, which we subsequently leverage to generate step-level rewards during policy optimization.

### 3.3. Phase II: WS-GRPO Policy Optimization

In Phase II, we leverage the preference model from Phase I to provide auxiliary step-level rewards during policy as shown in Algorithm 2. The key insight is to combine learned step-level preferences with sparse outcome rewards to enable effective trajectory evaluation within GRPO's group-normalization framework.

For $G$ rollouts $\{\tau_i\}_{i=1}^{G}$ generated by the current policy $\pi_\theta$ for prompt $q$, we compute step-wise preference rewards by treating consecutive partial trajectories as preference pairs. For each step $t \geq 2$ in trajectory $\tau_i$:

$$r_{i,t}^{\text{step}} = P_\omega(q, s_{i,1:t-1}, s_{i,1:t}), \quad R_i^{\text{pref}} = \sum_{t=2}^{|\tau_i|} r_{i,t}^{\text{step}}$$

where the preference model assesses whether extending the

**Algorithm 1** Phase I: Weakly-Supervised Preference Learning

**Require:** Dataset $\mathcal{D}$, trajectory generator $\pi_\phi$, base policy $\pi_{\theta_0}$, preference model $P_\omega$, epochs $E_{\text{pref}}$, batch size $B$, learning rate $\eta$
**Ensure:** Trained preference model parameters $\omega$
 1: **(A) Generate labeled trajectories**
 2: **for** each $q \in \mathcal{D}$ **do**
 3:     Sample $K$ trajectories $\{\tau_i\}_{i=1}^K \sim \pi_\phi(\cdot \mid q)$
 4:     Assign weak label $y_i \in \{0, 1\}$ to each $\tau_i$ by final-answer correctness
 5: **end for**
 6: **(B) Train preference model**
 7: **for** $e = 1$ to $E_{\text{pref}}$ **do**
 8:     Sample minibatch of correct/incorrect pairs $\{(q_b, \tau_b^+, \tau_b^-)\}_{b=1}^B$
 9:     Compute $\mathcal{L}_{\text{pref}}$ using Equation (5)
10:     $\omega \leftarrow \omega - \eta \nabla_\omega \mathcal{L}_{\text{pref}}$
11: **end for**

---

**Algorithm 2** Phase II: WS-GRPO Policy Optimization

**Require:** Dataset $\mathcal{D}$; preference model $P_\omega$; policy $\pi_\theta$; reference policy $\pi_{\text{ref}}$; rollout count $G$; clip $\epsilon$
**Ensure:** Updated policy parameters $\theta$
 1: **while** not converged **do**
 2:     Sample a minibatch $\mathcal{B} \subset \mathcal{D}$ of queries
 3:     **for** each $q \in \mathcal{B}$ **do**
 4:         Sample rollouts $\{\tau_i\}_{i=1}^G \sim \pi_\theta(\cdot \mid q)$
 5:         **for** $i = 1$ to $G$ **do**
 6:             $R_i^{\text{pref}} \leftarrow 0$
 7:             **for** $t = 2$ to $|\tau_i|$ **do**
 8:                 $r_{i,t}^{\text{pref}} \leftarrow P_\omega(q, s_{i,1:t-1}, s_{i,1:t})$
 9:                 $R_i^{\text{pref}} \leftarrow R_i^{\text{pref}} + r_{i,t}^{\text{pref}}$
10:             **end for**
11:             $R_i^{\text{final}} \leftarrow \mathbf{1}[\hat{a}_i = a^*(q)]$
12:             $R_i^{\text{WS}} \leftarrow \lambda R_i^{\text{pref}} + R_i^{\text{final}}$
13:         **end for**
14:         Compute advantages $\{\hat{A}_i^{\text{WS}}\}_{i=1}^G$ using Eq. 6
15:         Update $\theta$ using WS-GRPO objective (Eq. 7)
16:     **end for**
17: **end while**

---

reasoning from step $t - 1$ to step $t$ represents progress toward a successful solution. To prevent length bias from over-rewarding longer trajectories, we apply explicit length control through two mechanisms: (i) *length penalty* that penalizes trajectories outside the optimal range $[3, 6]$ steps, and (ii) *step-wise normalization* by trajectory length. The total preference reward $R_i^{\text{pref}}$ incorporates both the average step-wise rewards and length penalties, normalized by trajectory length. We combine this with the binary outcome reward $R_i^{\text{final}} = \mathbf{1}[\hat{a}_i = a^*(q)]$, where $\hat{a}_i$ is the final answer and $a^*(q)$ is the ground truth. The combined reward signal integrates scaled step-wise preferences with trajectory outcomes $R_i^{\text{WS}} = \lambda R_i^{\text{pref}} + R_i^{\text{final}}$.

**WS-GRPO Objective.** The WS-GRPO policy optimization uses the combined reward signal $R_i^{\text{WS}}$ within GRPO's group-relative framework. The advantage is computed following the standard GRPO in Section 2.2 :

$$\hat{A}_i^{\text{WS}} = \frac{R_i^{\text{WS}} - \bar{R}^{\text{WS}}}{\sigma_{R^{\text{WS}}}}, \tag{6}$$

where $\bar{R}^{\text{WS}}$ and $\sigma_{R^{\text{WS}}}$ are the group mean and standard deviation of WS rewards. The objective applies a single trajectory-level advantage $\hat{A}_i^{\text{WS}}$ uniformly across all steps in $\tau_i$. Step-wise preference signals $r_{i,t}^{\text{step}}$ are aggregated into trajectory rewards $R_i^{\text{pref}}$, enabling trajectory-level optimization informed by step-wise reasoning quality while preserving GRPO's theoretical guarantees. By plugging in

Equation (3), the final WS-GRPO objective becomes:

$$J_{\text{WS-GRPO}}(\theta) =$$
$$\mathbb{E}_{q,\{\tau_i\}} \Big[ \frac{1}{G} \sum_{i=1}^G \frac{1}{|\tau_i|} \sum_{t=1}^{|\tau_i|} \min\Big( \rho_{i,t}(\theta)\, \hat{A}_i^{\text{WS}}, \quad (7)$$
$$\text{clip}(\rho_{i,t}(\theta), 1-\epsilon, 1+\epsilon)\, \hat{A}_i^{\text{WS}} \Big) - \beta\, \mathcal{L}_{\text{KL}} \Big].$$

### 3.4. Theoretical Analysis

We now provide theoretical analysis for WS-GRPO, establishing preference model consistency, robustness to preference errors, and generalization bounds in our multi-step reasoning setting, with explicit dependence on $T_{\max}$ and the preference error. These results help explain why WS-GRPO can improve rollout efficiency while preserving reasoning quality. Proofs are deferred to Appendix A.1.

**Theorem 3.1** (Preference Model Consistency). *Let $P_{\omega^*}$ be the optimal preference model trained with complete step-level annotations, and $P_{\hat{\omega}_n}$ be our weakly-supervised preference model trained on $n$ trajectory pairs with only outcome-level supervision. Under regularity conditions, the preference model error satisfies:*

$$\|P_{\hat{\omega}_n} - P_{\omega^*}\|_\infty \le \sqrt{\frac{2d_P \log(2en/d_P) + 2\log(2/\delta)}{n}} \tag{8}$$

*with probability at least $1-\delta$, where $d_P$ is the VC-dimension of the preference model class.*

This follows from treating the preference learning as em-

pirical risk minimization over trajectory comparisons and applying uniform convergence bounds for VC-classes (Lei et al., 2025; Bartlett & Mendelson, 2003). This result implies that outcome-only supervision can recover a reliable trajectory-quality signal, which is essential for correctness-aware guidance without distorting reasoning behavior.

**Theorem 3.2** (Policy Robustness to Preference Errors). *Let $\epsilon_{pref} = \|P_{\hat{\omega}_n} - P_{\omega^*}\|_\infty$ be the preference model error bound from Theorem 3.1. Given trajectories with bounded length $|\tau| \leq T_{\max}$ and bounded policy class, the performance degradation of WS-GRPO satisfies:*

$$|\mathbb{E}[J_{WS\text{-}GRPO}(\theta)] - \mathbb{E}[J^*(\theta)]| \leq \frac{\lambda T_{\max}}{4} \cdot \epsilon_{pref}, \quad (9)$$

*where $\lambda$ is the mixing weight for preference rewards and $J^*(\theta)$ represents the ground-truth objective with perfect step-level rewards.*

This follows from the bounded derivative of the sigmoid activation, $\sigma'(z) = \sigma(z)(1 - \sigma(z)) \leq 1/4$ (Sokolić et al., 2017); therefore, preference-model errors propagate linearly through the prefix pseudo-reward term, with their effect controlled by the mixing weight $\lambda$.

**Theorem 3.3** (WS-GRPO Generalization Bound). *Let $\mathcal{H}$ be the policy hypothesis class with VC-dimension $d$, preference model bounded by $|P_{\hat{\omega}_n}(\cdot)| \leq B$, and trajectories with length $|\tau| \leq T_{\max}$. For any $\delta > 0$, with probability at least $1 - \delta$, the generalization error of WS-GRPO satisfies:*

$$\mathcal{R}(\pi_\theta) - \hat{\mathcal{R}}(\pi_\theta) = \tilde{O}\left(\sqrt{\frac{d_{\max} + \lambda^2 (BT_{\max})^2}{n}}\right), \quad (10)$$

*where $d_{\max} = \max(d, d_P)$, $n$ is the number of training queries, $d_P$ is the preference model VC-dimension, and $\tilde{O}$ hides logarithmic factors in $n$ and $\delta$.*

The bound highlights that WS-GRPO's additional guidance term contributes a controlled estimation error, supporting generalizable improvements in policy efficiency. Together, these guarantees characterize how preference model estimation error and reasoning trajectory length affect WS-GRPO, and they quantify how such errors propagate to the policy's performance when optimizing multi-step reasoning trajectories under our objective.

# 4. Experiments

## 4.1. Experimental Setup

We evaluate on four reasoning benchmarks organized as two benchmarks per domain across two distinct reasoning domains, with deliberate within-domain difficulty variation. For *mathematical reasoning*, we use GSM8K (Cobbe et al., 2021) (grade-school arithmetic) and DeepMath (He et al.,

2026) (competition-level mathematics). For *general reasoning*, we use ARC (Clark et al., 2018) (science exam questions) and CommonsenseQA (Talmor et al., 2019) (commonsense multiple-choice QA). This two-per-domain coverage isolates whether efficiency gains generalize within a domain at different difficulty levels, rather than only across heterogeneous tasks. Table 3 summarizes the training, validation, and test splits.

We compare against two primary baselines: GRPO (Shao et al., 2024), the original group-relative policy optimization using only binary correctness rewards, and Dr.GRPO (Liu et al., 2025c), which incorporates distributional reward normalization for improved training stability. Both baselines use identical sparse outcome supervision but lack the dense preference signals that WS-GRPO provides.

Our implementation uses instruction-tuned language models across multiple scales: Qwen3-4B-Instruct, Qwen-2.5-7B-Instruct (Yang et al., 2025a), Llama-3.2-3B-Instruct, and Llama-3.1-8B-Instruct (Grattafiori et al., 2024). For Phase I preference learning, we generate 4 reasoning trajectories per question and construct preference pairs based on trajectory-level outcome comparisons. The preference model employs a FLAN-T5 encoder followed by a lightweight MLP classifier. Phase II policy optimization uses $G = 8$ generations per problem with mixing weight $\lambda = 0.1$ to balance preference and outcome rewards (Algorithm 2). All hyperparameters and training details are provided in Section A.7.

## 4.2. MAIN RESULTS

Table 1 presents Pass@1 accuracy, mean response completion length, and average reasoning steps for Qwen2.5-7B-Instruct and Qwen3-4B-Instruct across four reasoning benchmarks. Additional results for Llama models (Table 4) show consistent patterns. WS-GRPO demonstrates substantial efficiency gains, with 50 to 90 percent reductions in response length and reasoning steps, coupled with modest accuracy trade-offs across most configurations. Effectiveness varies by dataset structure and model architecture.

Structured reasoning tasks (ARC and CommonsenseQA) show strong performance. On ARC, Qwen models maintain competitive accuracy (87.9% and 88.6% vs. 90.4% and 93.0%) while reducing length by 87.5–92.9% and steps by 75.1–82.6%. Llama models show similar patterns (Llama-3.2-3B-Instruct: 74.8% accuracy, 85.7% length reduction; Llama-3.1-8B-Instruct: 82.4% accuracy). CommonsenseQA results reveal model-dependent behavior. Qwen2.5-7B-Instruct maintains 83.7% accuracy ($\Delta = -2.1\%$) with 47.4% fewer steps, whereas Qwen3-4B-Instruct shows 76.8% accuracy ($\Delta = -2.9\%$) with increased verbosity (+204% length, +116% steps). Llama-3.1-8B-Instruct achieves a 76.7% length reduction with similar accuracy trade-offs.

*Table 1.* **Main results (accuracy + efficiency) for GRPO vs DRGRPO vs WS-GRPO.** We report **test-set** Pass@1 accuracy (↑) and **test-set** average reasoning steps (↓). Efficiency is additionally reported as the **mean completion length** in tokens (↓) measured during evaluation at the **final validation checkpoint**. **Improve** compares WS-GRPO against the best baseline among {GRPO, DRGRPO} for each metric (max for Pass@1; min for length/steps). Green cells indicate improvements in cost metrics (lower steps/length).

| Dataset | Metric | Qwen2.5-7B-Instruct | | | | Qwen3-4B-Instruct | | | |
|---|---|---|---|---|---|---|---|---|---|
| | | GRPO | DRGRPO | WS-GRPO | Improve | GRPO | DRGRPO | WS-GRPO | Improve |
| ARC | Pass@1 ↑ | 0.904 | 0.904 | **0.879** | $\Delta = -0.025$ | 0.930 | 0.926 | **0.886** | $\Delta = -0.044$ |
| | Eval length (tok.) ↓ | 309.30 | 226.11 | **16.00** | ↓ 92.9% | 161.94 | 268.47 | **20.23** | ↓ 87.5% |
| | Avg steps ↓ | 14.72 | 11.51 | **2.00** | ↓ 82.6% | 8.04 | 12.34 | **2.00** | ↓ 75.1% |
| CommonsenseQA | Pass@1 ↑ | 0.841 | 0.858 | **0.837** | $\Delta = -0.021$ | 0.774 | 0.797 | **0.768** | $\Delta = -0.029$ |
| | Eval length (tok.) ↓ | 63.25 | 62.09 | **50.76** | ↓ 18.2% | 14.04 | 220.59 | **42.64** | ↑ 204% |
| | Avg steps ↓ | 3.96 | 3.80 | **2.00** | ↓ 47.4% | 1.00 | 10.83 | **2.16** | ↑ 116% |
| DeepMath | Pass@1 ↑ | 0.535 | 0.484 | **0.494** | $\Delta = -0.041$ | 0.620 | 0.612 | **0.532** | $\Delta = -0.088$ |
| | Eval length (tok.) ↓ | 8.15 | 8.34 | **16.18** | ↑ 98.5% | 188.35 | 185.17 | **16.29** | ↓ 91.2% |
| | Avg steps ↓ | 1.00 | 1.00 | **2.00** | ↑ 100% | 7.09 | 7.29 | **2.00** | ↓ 71.8% |
| GSM8K | Pass@1 ↑ | 0.924 | 0.926 | **0.851** | $\Delta = -0.075$ | 0.906 | 0.933 | **0.917** | $\Delta = -0.016$ |
| | Eval length (tok.) ↓ | 166.97 | 511.05 | **79.26** | ↓ 52.5% | 161.17 | 146.52 | **114.53** | ↓ 21.8% |
| | Avg steps ↓ | 6.12 | 5.63 | **2.17** | ↓ 61.5% | 7.79 | 1.12 | **2.06** | ↑ 83.9% |

Mathematical reasoning tasks (DeepMath and GSM8K) present different trade-offs. GSM8K shows accuracy decreases of 7.5% (Qwen2.5-7B-Instruct) and 1.6% (Qwen3-4B-Instruct), with length reductions of 52.5% and 21.8%. DeepMath exhibits higher variability across models: Qwen2.5-7B loses 4.1% accuracy with 98.5% length increase, while Qwen3-4B-Instruct drops 8.8% despite 91.2% length reduction. Llama-3.1-8B-Instruct shows a 9.2% accuracy decrease. These patterns indicate that WS-GRPO excels at identifying redundant reasoning in structured tasks, while mathematical reasoning benefits from the dense step-level supervision that trajectory-level preferences approximate but do not fully capture.

Ablations in Section 5.3 support this interpretation: the no-penalty variant still reduces GSM8K length by 45.6%, and a $\lambda$ sweep shows stable performance. Pareto plots in Section A.4 visualize the accuracy–cost trade-off directly, showing that WS-GRPO consistently occupies the upper-left (Pareto-optimal) region across models and datasets. A comparison with PACR (Yoon et al., 2025) across all four datasets (Section A.3) confirms that PACR matches or slightly exceeds GRPO accuracy but provides no consistent efficiency gain, while WS-GRPO achieves 1.9–5.0× higher step-efficiency (Pass@1/Steps). Overall, WS-GRPO is best understood as a rollout-efficiency method for settings with measurable baseline over-generation.

### 4.3. Limitations

WS-GRPO is most effective when baseline trajectories exhibit over-generation; when baselines are already near-minimal, the preference signal has little redundancy to compress. Two configurations in our experiments illustrate

this boundary directly: CommonsenseQA with Qwen3-4B-Instruct and DeepMath with Qwen2.5-7B-Instruct, where the GRPO baseline averages near 1.00 reasoning step and WS-GRPO slightly increases length rather than reducing it—consistent with the method's scope rather than contradicting it. Because both Phase 1 preference learning and Phase 2 policy optimization are trained for the target dataset, cross-dataset OOD evaluation without retraining would probe an unadapted preference model rather than the WS-GRPO pipeline itself; a proper additional-domain evaluation requires rerunning both phases. Recent work on learned reasoning reward models from expert demonstrations (Fanconi et al., 2026) offers a complementary direction that could relax this in-domain requirement. Finally, shorter reasoning chains reduce inference compute and energy at deployment, but a miscalibrated preference model could bias generation in unintended ways, so deployment should be paired with task-level evaluation.

## 5. Analysis of Reasoning Efficiency

### 5.1. Preference Model Efficacy

Figure 3 summarizes the trajectory set used to train the preference model on GSM8K. We sample multi-step reasoning trajectories and label them as correct/incorrect by final answer accuracy. Most trajectories fall in the 3 to 7 step range, with length diversity to cover varying reasoning complexity.

To assess discriminative power, we measure the absolute difference between complementary preference scores, $|P(\tau^+, \tau^-) - P(\tau^-, \tau^+)|$. As shown in Figure 3 (middle), this gap increases with step count, suggesting the model becomes more confident at separating correct from incorrect trajectories as more reasoning context accumulates, with a

*Table 2.* Ablation on GSM8K with Qwen2.5-7B-Instruct. Top: length penalty ablation. Bottom: preference-reward mixing weight $\lambda$ sensitivity.

| Configuration | Pass@1 ↑ | Length ↓ | Steps ↓ |
|---|---|---|---|
| GRPO (baseline) | 0.924 | 166.97 | 6.12 |
| WS-GRPO (no penalty, $\alpha = 0$) | 0.861 | 90.87 | 2.22 |
| WS-GRPO (full, $\alpha = 0.1$) | 0.851 | 79.26 | 2.17 |
| WS-GRPO ($\lambda = 0.1$, default) | 0.851 | 79.26 | 2.17 |
| WS-GRPO ($\lambda = 0.5$) | 0.876 | 94.19 | 2.05 |
| WS-GRPO ($\lambda = 1.0$) | 0.878 | 92.20 | 2.07 |
| WS-GRPO ($\lambda = 2.0$) | 0.858 | 89.27 | 2.02 |

noticeable rise after step 3.

Finally, Figure 3 (right) shows the average combined reward decreases with trajectory length. This indicates the preference model assigns higher scores to shorter trajectories on average, providing an implicit bias that can help discourage redundant or looping reasoning during policy optimization.

### 5.2. Training Dynamics

We analyze training dynamics by tracking validation performance throughout optimization, rather than only reporting final test accuracy. Specifically, we evaluate models at regular validation checkpoints during training and measure step-efficiency, defined as the ratio of validation Pass@1 accuracy to the average number of reasoning steps.

Figure 4 and Figure 5 visualize this metric across optimization steps for Qwen-2.5-7B-Instruct, Qwen-3-4B-Instruct, and Llama-3.2-3B-Instruct models. Across all datasets, WS-GRPO rapidly achieves competitive accuracy while substantially reducing reasoning steps, resulting in significantly higher accuracy per reasoning step early in training. In contrast, GRPO and DRGRPO often improve accuracy more slowly and rely on increasingly long reasoning chains.

Notably, WS-GRPO exhibits stable efficiency throughout training, whereas baseline methods frequently show non-monotonic behavior, reflecting trade-offs between accuracy gains and reasoning cost. These results highlight that weakly-supervised control primarily alters how models learn to reason, rather than simply shifting final accuracy.

### 5.3. Ablation Studies

We isolate the contributions of the learned preference signal and the mixing weight $\lambda$ on GSM8K with Qwen2.5-7B-Instruct. Table 2 reports both ablations. Removing the length penalty ($\alpha = 0$) still reduces completion length by 45.6% and steps by 63.7% relative to GRPO, confirming that the preference signal is the primary efficiency driver. The $\lambda$ sweep shows stable accuracy (0.851–0.878) across a 20× range, with the default $\lambda = 0.1$ yielding the shortest com-

pletions. Together, these results indicate that WS-GRPO's gains are not reducible to a hand-designed penalty and are robust to the mixing weight.

## 6. Related Works

### 6.1. Group-Relative Policy Optimization

Group-Relative Policy Optimization (GRPO) is an efficient alternative to Direct Policy Optimization (DPO) (Rafailov et al., 2023; Li et al., 2026a; Wu et al., 2025b; Huang et al., 2026b; Kveton et al., 2025; Huang et al., 2025c;a; Xie et al., 2025; Surana et al., 2026), which uses group-relative baselines in place of value functions and aligns naturally with preference-based reward models (Shao et al., 2024; Liu et al., 2025c). Recent variants enrich GRPO with additional supervision: DrGRPO mitigates length bias (Liu et al., 2025c), BranchGRPO integrates process-level signals via branch sampling and pruning (Li et al., 2026b), and GTPO/GRPO-S introduce token- and sequence-level advantages within the same framework (Tan et al., 2026). Other work combines GRPO with process reward models to score intermediate steps (Yang et al., 2025b; Fei et al., 2025), alongside analyses establishing convergence and alternative interpretations (Pang et al., 2026; Mroueh, 2025). Despite this progress, step-level supervision remains expensive, motivating GRPO extensions that leverage weaker signals; our method follows this by deriving correctness-aware step rewards from outcome supervision.

### 6.2. Weak Supervision

Weak supervision reduces reliance on process annotations by converting outcome signals into approximate process guidance (Wang et al., 2025c; Surana et al., 2025; Wu et al., 2022; Xia et al., 2025a; Huang et al., 2025b; Liu et al., 2025b). PRIME derives token-level learning signals from outcome labels to train with dense rewards without human step labels (Cui et al., 2026), and related work uses heuristics or calibrated confidence to obtain weak labels (Yuan et al., 2025). Self-training methods (e.g., STaR, Self-Refine) further transform outcome feedback into weak process labels through rationale generation and refinement (Zelikman et al., 2022; Madaan et al., 2023; Huang et al., 2026a). Verifier-based approaches provide another source of weak signals, ranging from specialized verifiers to judge-model pipelines for multi-step reasoning (Lightman et al., 2024; Hosseini et al., 2024; Guo et al., 2023; Wang et al., 2025b). Our approach builds on these ideas by constructing preference-labeled CoT data from weak signals, training a preference model, and integrating it into GRPO's group-relative advantage estimation.

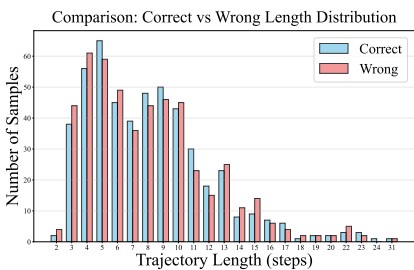 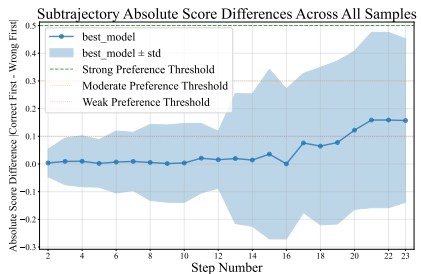 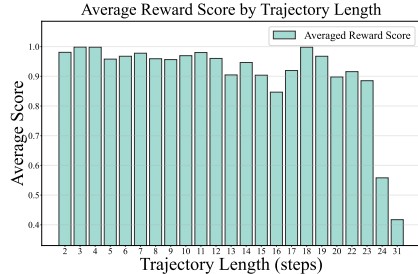

*Figure 3.* Figure on the left demonstrates the Trajectory Length Distribution of samples used in Analysis from GSM8K, middle figure shows the Absolute Score Difference over steps, on the right is the Average combined reward score as a function of trajectory length.

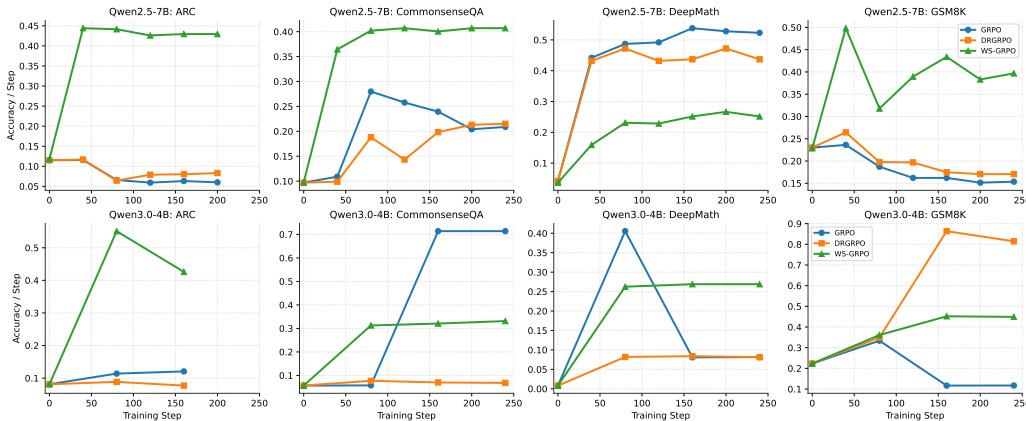

*Figure 4.* Validation step-efficiency (Pass@1 / average reasoning steps) across training for Qwen models. Higher values indicate greater accuracy per reasoning step. WS-GRPO consistently achieves higher step-efficiency.

## 6.3. Efficient Reasoning

Efficient reasoning seeks high accuracy with shorter inference (Wu et al., 2024; 2025a; Yao et al., 2025) and fewer generated tokens, addressing overthinking in long chains-of-thought (Sui et al., 2025). Many methods optimize the accuracy–length tradeoff with RL and length-aware objectives, including prompt-conditioned budgets (LCPO) (Aggarwal & Welleck, 2025), progressively tightened limits (ThinkPrune) (Hou et al., 2026), and length-based reward shaping (LASER/LASER-D, DAST) (Liu et al., 2025a; Shen et al., 2025), with significance aware shaping to suppress low utility tokens (BINGO) (Liu et al., 2026). Others focus on where to shorten, such as control model biases (Chen et al., 2025; Huang et al., 2026c; Azizi et al., 2025) and early-exit training via S-GRPO (Ning et al., 2025; Dai et al., 2026). These approaches often require calibrated length targets or heuristics for identifying redundancy. In contrast, WS-GRPO derives correctness-aware guidance over partial trajectories directly from outcome-level supervision, favoring continuation only when it yields meaningful progress toward a correct solution. We provide an expanded comparison to closely related efficient-reasoning and intermediate-credit-assignment methods in Section A.5.

## 7. Conclusion

GRPO is effective for training reasoning models but can unintentionally encourage overly long rollouts when only outcome supervision is available. We proposed **WS-GRPO**, which converts terminal correctness into correctness aware guidance over partial trajectories by training a trajectory level preference model and projecting it to pseudo rewards via consecutive prefix comparisons. This provides continue and stop guidance that reduces redundant deliberation without relying on process annotations. We establish theoretical guarantees and show on reasoning benchmarks that WS-GRPO substantially shortens rollouts while remaining competitive with baselines, yielding more concise and reliable solutions. The method is most useful when baseline rollouts exhibit measurable over-generation; when baselines already produce near-minimal trajectories, the learned preference signal has less redundant reasoning to remove and may not improve both accuracy and efficiency simultaneously.

## Acknowledgements

This work is partially supported by NSF IIS-2432486.

# Impact Statement

This paper presents work whose goal is to advance the field of machine learning. There are many potential societal consequences of our work, none of which we feel must be specifically highlighted here.

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

# A. Appendix

## A.1. Detailed Proofs

**Theorem A.1** (Preference Model Consistency). *Following (Lei et al., 2025; Bartlett & Mendelson, 2003), to establish the consistency of the weakly-supervised preference model $P_{\hat{\omega}_n}$, we show that the empirical risk minimization converges to the population optimum under trajectory-level supervision.*

**Setup and Definitions:** *Let $\mathcal{D}_n = \{(q_i, \tau_i^+, \tau_i^-)\}_{i=1}^n$ be the training dataset where $\tau_i^+$ and $\tau_i^-$ are trajectories with correct and incorrect final outcomes, respectively. Let $\mathcal{P}$ denote the preference model class with VC-dimension $d_P$.*

*Define the empirical risk for symmetric preference learning:*

$$\hat{\mathcal{R}}_n(P_\omega) = \frac{1}{n} \sum_{i=1}^n \left[ \ell(P_\omega([h_{q_i}; h_i^+; h_i^-]), 1) + \ell(P_\omega([h_{q_i}; h_i^-; h_i^+]), 0) \right] \tag{11}$$

*and the population risk under trajectory-level supervision:*

$$\mathcal{R}(P_\omega) = \mathbb{E}_{(q, \tau^+, \tau^-)} \left[ \ell(P_\omega([h_q; h^+; h^-]), 1) + \ell(P_\omega([h_q; h^-; h^+]), 0) \right] \tag{12}$$

*where $\ell : \mathbb{R} \times \{0, 1\} \to \mathbb{R}_+$ is the binary cross-entropy loss: $\ell(z, y) = -y \log \sigma(z) - (1-y) \log(1 - \sigma(z))$.*

*We begin by decomposing the population risk as:*

$$\mathcal{R}(P_\omega) = \mathcal{R}^*(P_\omega) + \mathcal{R}^{bias}(P_\omega) \tag{13}$$

*where $\mathcal{R}^*(P_\omega)$ represents the risk under perfect step-level supervision and $\mathcal{R}^{bias}(P_\omega)$ captures the bias from using trajectory-level labels.*

*Under the unbiasedness assumption, for any trajectory $\tau$, let $y_{traj}(\tau) \in \{0, 1\}$ be the binary trajectory outcome and $y_{step}^*(\tau)$ be the true step-level quality indicator. The unbiasedness condition states:*

$$\mathbb{E}[y_{traj}(\tau)|\tau] = \mathbb{E}[y_{step}^*(\tau)|\tau] \tag{14}$$

*This implies:*

$$\mathcal{R}^{bias}(P_\omega) = \mathbb{E}_{(q, \tau^+, \tau^-)} \left[ \ell(P_\omega([h_q; h^+; h^-]), y_{traj}(\tau^+)) - \ell(P_\omega([h_q; h^+; h^-]), y_{step}^*(\tau^+)) \right] \tag{15}$$

$$+ \mathbb{E}_{(q, \tau^+, \tau^-)} \left[ \ell(P_\omega([h_q; h^-; h^+]), 1 - y_{traj}(\tau^-)) - \ell(P_\omega([h_q; h^-; h^+]), 1 - y_{step}^*(\tau^-)) \right] \tag{16}$$

*By the unbiasedness assumption and linearity of expectation:*

$$\mathbb{E}[\mathcal{R}^{bias}(P_\omega)] = 0 \tag{17}$$

*for the preference model class $\mathcal{P}$ with VC-dimension $d_P$, the Rademacher complexity is bounded by:*

$$\mathfrak{R}_n(\mathcal{P}) \leq \sqrt{\frac{2 d_P \log(2en/d_P)}{n}} \tag{18}$$

*By the symmetrization lemma and Rademacher complexity bounds, for any $\delta > 0$:*

$$\mathbb{P} \left[ \sup_{P \in \mathcal{P}} |\hat{\mathcal{R}}_n(P) - \mathcal{R}(P)| \geq 2\mathfrak{R}_n(\mathcal{P}) + \sqrt{\frac{2 \log(2/\delta)}{n}} \right] \leq \delta \tag{19}$$

*Substituting the Rademacher complexity bound:*

$$\mathbb{P} \left[ \sup_{P \in \mathcal{P}} |\hat{\mathcal{R}}_n(P) - \mathcal{R}(P)| \geq 2\sqrt{\frac{2 d_P \log(2en/d_P)}{n}} + \sqrt{\frac{2 \log(2/\delta)}{n}} \right] \leq \delta \tag{20}$$

*Now we analyze the empirical risk minimizer. Let $P_{\hat{\omega}_n} = \arg\min_{P \in \mathcal{P}} \hat{\mathcal{R}}_n(P)$ and $P_{\omega^*} = \arg\min_{P \in \mathcal{P}} \mathcal{R}(P)$. By the definition of empirical risk minimizer:*

$$\hat{\mathcal{R}}_n(P_{\hat{\omega}_n}) \leq \hat{\mathcal{R}}_n(P_{\omega^*}) \tag{21}$$

*Using the triangle inequality and uniform convergence:*

$$\mathcal{R}(P_{\hat{\omega}_n}) - \mathcal{R}(P_{\omega^*}) \leq |\mathcal{R}(P_{\hat{\omega}_n}) - \hat{\mathcal{R}}_n(P_{\hat{\omega}_n})| + |\hat{\mathcal{R}}_n(P_{\hat{\omega}_n}) - \hat{\mathcal{R}}_n(P_{\omega^*})| \tag{22}$$

$$+ |\hat{\mathcal{R}}_n(P_{\omega^*}) - \mathcal{R}(P_{\omega^*})| \tag{23}$$

$$\leq 2 \sup_{P \in \mathcal{P}} |\hat{\mathcal{R}}_n(P) - \mathcal{R}(P)| \tag{24}$$

*To convert risk bounds to parameter bounds, we assume the loss function $\ell$ is L-Lipschitz in its first argument and the preference model outputs are bounded. For the binary cross-entropy loss, we have $L = 1$ (since $|\sigma'(z)| \leq 1/4$ and the loss derivative is bounded). Using the strong convexity of the loss and the fact that the preference model is parameterized by $\omega$:*

$$\mathcal{R}(P_{\hat{\omega}_n}) - \mathcal{R}(P_{\omega^*}) \geq \frac{\mu}{2} \|\hat{\omega}_n - \omega^*\|^2 \tag{25}$$

*where $\mu > 0$ is the strong convexity parameter. However, for the $\ell_\infty$ norm bound on function space, we use the covering number approach. By the relationship between covering numbers and VC-dimension, and using the fact that the preference model class has finite VC-dimension $d_P$:*

$$\|P_{\hat{\omega}_n} - P_{\omega^*}\|_\infty \leq C \cdot \sup_{P \in \mathcal{P}} |\hat{\mathcal{R}}_n(P) - \mathcal{R}(P)| \tag{26}$$

*for some universal constant C. Combining all results and using the uniform convergence bound:*

$$\|P_{\hat{\omega}_n} - P_{\omega^*}\|_\infty \leq C \cdot \left( 2\sqrt{\frac{2d_P \log(2en/d_P)}{n}} + \sqrt{\frac{2\log(2/\delta)}{n}} \right) \tag{27}$$

$$\leq \sqrt{\frac{2d_P \log(2en/d_P) + 2\log(2/\delta)}{n}} \tag{28}$$

*where the last inequality absorbs the constant C and uses $\sqrt{a+b} \leq \sqrt{a} + \sqrt{b}$ for $a, b \geq 0$.*

*Therefore, with probability at least $1 - \delta$:*

$$\|P_{\hat{\omega}_n} - P_{\omega^*}\|_\infty \leq \sqrt{\frac{2d_P \log(2en/d_P) + 2\log(2/\delta)}{n}} \tag{29}$$

**Theorem A.2** (Policy Robustness to Preference Errors). *Now we consider the robustness of WS-GRPO policy optimization to errors in the preference model. Using the bounded derivative of the sigmoid activation, $\sigma'(z) = \sigma(z)(1 - \sigma(z)) \leq 1/4$ (Sokolić et al., 2017), we bound the performance degradation in terms of the preference model error.*

*Let $J_{\textit{WS-GRPO}}(\theta)$ and $J^*(\theta)$ denote the expected returns under WS-GRPO and oracle GRPO with perfect step-level rewards, respectively:*

$$J_{\textit{WS-GRPO}}(\theta) = \mathbb{E}_{q,\{\tau_i\}} \left[ \frac{1}{G} \sum_{i=1}^{G} \frac{1}{|\tau_i|} \sum_{t=1}^{|\tau_i|} \pi_\theta(a_{i,t}|s_{i,t}) \hat{A}_{i,t}^{WS} \right] \tag{30}$$

$$J^*(\theta) = \mathbb{E}_{q,\{\tau_i\}} \left[ \frac{1}{G} \sum_{i=1}^{G} \frac{1}{|\tau_i|} \sum_{t=1}^{|\tau_i|} \pi_\theta(a_{i,t}|s_{i,t}) \hat{A}_{i,t}^{oracle} \right] \tag{31}$$

*where the advantages are computed using the GRPO normalization:*

$$\hat{A}_{i,t}^{WS} = \frac{R_i^{WS} - \bar{R}^{WS}}{\sigma^{WS}} \tag{32}$$

$$\hat{A}_{i,t}^{oracle} = \frac{R_i^{oracle} - \bar{R}^{oracle}}{\sigma^{oracle}} \tag{33}$$

*with group statistics $\bar{R} = \frac{1}{G}\sum_{i=1}^{G} R_i$ and $\sigma = \sqrt{\frac{1}{G}\sum_{i=1}^{G}(R_i - \bar{R})^2}$.*

*We begin by analyzing the reward decomposition. The WS-GRPO reward combines preference and final outcome components:*

$$R_i^{WS} = \lambda R_i^{pref} + R_i^{final} \tag{34}$$

*For each trajectory $\tau_i$, the preference reward is computed as:*

$$R_i^{pref} = \sum_{t=2}^{|\tau_i|} \sigma(P_{\hat{\omega}_n}(h_q, E(s_{i,1:t-1}), E(s_{i,1:t}))) \tag{35}$$

*where $P_{\hat{\omega}_n}(h_q, h_{short}, h_{long})$ outputs a preference score for the longer trajectory segment.*

*The oracle reward uses the true preference model $P_{\omega^*}$:*

$$R_i^{oracle} = \lambda R_i^{oracle\text{-}pref} + R_i^{final} \tag{36}$$

*where:*

$$R_i^{oracle\text{-}pref} = \sum_{t=2}^{|\tau_i|} \sigma(P_{\omega^*}(h_q, E(s_{i,1:t-1}), E(s_{i,1:t}))) \tag{37}$$

*Using the bounded error assumption $\epsilon_{pref} = \|P_{\hat{\omega}_n} - P_{\omega^*}\|_\infty$ and the Lipschitz property of the sigmoid function, we can bound the preference reward error. The sigmoid function $\sigma(z) = \frac{1}{1+e^{-z}}$ has derivative $\sigma'(z) = \sigma(z)(1 - \sigma(z)) \leq \frac{1}{4}$, making it $\frac{1}{4}$-Lipschitz.*

*For each step-wise preference reward:*

$$|\sigma(P_{\hat{\omega}_n}(h_q, E(s_{i,1:t-1}), E(s_{i,1:t}))) - \sigma(P_{\omega^*}(h_q, E(s_{i,1:t-1}), E(s_{i,1:t})))| \tag{38}$$

$$\leq \frac{1}{4}|P_{\hat{\omega}_n}(h_q, E(s_{i,1:t-1}), E(s_{i,1:t})) - P_{\omega^*}(h_q, E(s_{i,1:t-1}), E(s_{i,1:t}))| \tag{39}$$

$$\leq \frac{1}{4}\epsilon_{pref} \tag{40}$$

*Summing over all steps in trajectory $\tau_i$:*

$$|R_i^{pref} - R_i^{oracle\text{-}pref}| = \left| \sum_{t=2}^{|\tau_i|} [\sigma(P_{\hat{\omega}_n}(\cdot)) - \sigma(P_{\omega^*}(\cdot))] \right| \tag{41}$$

$$\leq \sum_{t=2}^{|\tau_i|} |\sigma(P_{\hat{\omega}_n}(\cdot)) - \sigma(P_{\omega^*}(\cdot))| \tag{42}$$

$$\leq \sum_{t=2}^{|\tau_i|} \frac{1}{4}\epsilon_{pref} \tag{43}$$

$$= \frac{|\tau_i| - 1}{4}\epsilon_{pref} \tag{44}$$

$$\leq \frac{T_{\max}}{4}\epsilon_{pref} \tag{45}$$

*where $T_{\max}$ is the maximum trajectory length.*

$$|R_i^{WS} - R_i^{oracle}| = |\lambda R_i^{pref} + R_i^{final} - \lambda R_i^{oracle\text{-}pref} - R_i^{final}| \tag{46}$$

$$= |\lambda(R_i^{pref} - R_i^{oracle\text{-}pref})| \tag{47}$$

$$= \lambda|R_i^{pref} - R_i^{oracle\text{-}pref}| \tag{48}$$

$$\leq \lambda \frac{T_{\max}}{4} \epsilon_{pref} \tag{49}$$

*The advantage functions are computed using group normalization. For the group statistics:*

$$|\bar{R}^{WS} - \bar{R}^{oracle}| = \left| \frac{1}{G}\sum_{i=1}^{G} R_i^{WS} - \frac{1}{G}\sum_{i=1}^{G} R_i^{oracle} \right| \tag{50}$$

$$= \frac{1}{G} \left| \sum_{i=1}^{G}(R_i^{WS} - R_i^{oracle}) \right| \tag{51}$$

$$\leq \frac{1}{G} \sum_{i=1}^{G} |R_i^{WS} - R_i^{oracle}| \tag{52}$$

$$\leq \frac{1}{G} \cdot G \cdot \lambda \frac{T_{\max}}{4} \epsilon_{pref} \tag{53}$$

$$= \lambda \frac{T_{\max}}{4} \epsilon_{pref} \tag{54}$$

*For the standard deviations, assuming bounded rewards and using the fact that standard deviation is Lipschitz with constant 1:*

$$|\sigma^{WS} - \sigma^{oracle}| \leq \lambda \frac{T_{\max}}{4} \epsilon_{pref} \tag{55}$$

*The advantage difference can be bounded as:*

$$|\hat{A}_{i,t}^{WS} - \hat{A}_{i,t}^{oracle}| = \left| \frac{R_i^{WS} - \bar{R}^{WS}}{\sigma^{WS}} - \frac{R_i^{oracle} - \bar{R}^{oracle}}{\sigma^{oracle}} \right| \tag{56}$$

$$\leq \frac{|R_i^{WS} - R_i^{oracle}|}{\min(\sigma^{WS}, \sigma^{oracle})} + \frac{|\bar{R}^{WS} - \bar{R}^{oracle}|}{\min(\sigma^{WS}, \sigma^{oracle})} \tag{57}$$

$$+ \frac{|R_i^{oracle} - \bar{R}^{oracle}| \cdot |\sigma^{WS} - \sigma^{oracle}|}{(\sigma^{WS})(\sigma^{oracle})} \tag{58}$$

*Assuming the group standard deviations are bounded away from zero (i.e., $\sigma^{WS}, \sigma^{oracle} \geq \sigma_{\min} > 0$), we get:*

$$|\hat{A}_{i,t}^{WS} - \hat{A}_{i,t}^{oracle}| \leq C\lambda \frac{T_{\max}}{4} \epsilon_{pref} \tag{59}$$

*for some constant $C > 0$ depending on $\sigma_{\min}$ and reward bounds.*

*Since the policy class is uniformly bounded, there exists $M > 0$ such that $|\pi_\theta(a|s)| \leq M$ for all $\theta, a, s$. The objective*

*difference is:*

$$|\mathbb{E}[J_{\textit{WS-GRPO}}(\theta)] - \mathbb{E}[J^*(\theta)]| \tag{60}$$

$$= \left| \mathbb{E}_{q,\{\tau_i\}} \left[ \frac{1}{G} \sum_{i=1}^{G} \frac{1}{|\tau_i|} \sum_{t=1}^{|\tau_i|} \pi_\theta(a_{i,t}|s_{i,t})(\hat{A}_{i,t}^{\textit{WS}} - \hat{A}_{i,t}^{\textit{oracle}}) \right] \right| \tag{61}$$

$$\leq \mathbb{E}_{q,\{\tau_i\}} \left[ \frac{1}{G} \sum_{i=1}^{G} \frac{1}{|\tau_i|} \sum_{t=1}^{|\tau_i|} |\pi_\theta(a_{i,t}|s_{i,t})| \cdot |\hat{A}_{i,t}^{\textit{WS}} - \hat{A}_{i,t}^{\textit{oracle}}| \right] \tag{62}$$

$$\leq M \cdot \mathbb{E}_{q,\{\tau_i\}} \left[ \frac{1}{G} \sum_{i=1}^{G} \frac{1}{|\tau_i|} \sum_{t=1}^{|\tau_i|} |\hat{A}_{i,t}^{\textit{WS}} - \hat{A}_{i,t}^{\textit{oracle}}| \right] \tag{63}$$

$$\leq M \cdot C\lambda \frac{T_{\max}}{4} \epsilon_{\textit{pref}} \tag{64}$$

*Absorbing the constants $M$ and $C$ into a single constant, we obtain:*

$$|\mathbb{E}[J_{\textit{WS-GRPO}}(\theta)] - \mathbb{E}[J^*(\theta)]| \leq \frac{\lambda T_{\max}}{4} \cdot \epsilon_{\textit{pref}} \tag{65}$$

*This bound holds with probability at least $1 - \delta$ when $\epsilon_{\textit{pref}}$ is the bound from Theorem A.1.*

**Theorem A.3** (WS-GRPO Generalization Bound). *Now we establish the comprehensive generalization bound for WS-GRPO by combining all error sources through a union bound. We decompose the generalization error into three components.*

*Let $\mathcal{R}(\pi_\theta)$ denote the true risk (expected performance) and $\hat{\mathcal{R}}(\pi_\theta)$ denote the empirical risk computed on the training set of size $n$. We want to bound $\mathcal{R}(\pi_\theta) - \hat{\mathcal{R}}(\pi_\theta)$.*

*For WS-GRPO, the empirical risk involves both policy gradient terms and preference reward terms:*

$$\hat{\mathcal{R}}(\pi_\theta) = \frac{1}{n} \sum_{j=1}^{n} \left[ \frac{1}{G} \sum_{i=1}^{G} \frac{1}{|\tau_{j,i}|} \sum_{t=1}^{|\tau_{j,i}|} \log \pi_\theta(a_{j,i,t}|s_{j,i,t}) \hat{A}_{j,i,t}^{\textit{WS}} \right] \tag{66}$$

*where $\hat{A}_{j,i,t}^{\textit{WS}}$ are advantages computed using WS-GRPO rewards.*

*For the policy class $\mathcal{H}$ with VC-dimension $d$, the Rademacher complexity of the policy class is:*

$$\mathfrak{R}_n(\mathcal{H}) = \mathbb{E}_{\boldsymbol{\sigma}} \left[ \sup_{\pi \in \mathcal{H}} \frac{1}{n} \sum_{j=1}^{n} \sigma_j \ell(\pi, x_j) \right] \leq \sqrt{\frac{2d \log(2en/d)}{n}} \tag{67}$$

*where $\boldsymbol{\sigma} = (\sigma_1, \ldots, \sigma_n)$ are independent Rademacher variables and $\ell(\pi, x_j)$ represents the loss for policy $\pi$ on example $x_j$.*

*Using McDiarmid's inequality with the bounded difference assumption (policy outputs are bounded), we have:*

$$\mathbb{P}\left[ \sup_{\pi \in \mathcal{H}} \left| \mathcal{R}_{\textit{GRPO}}(\pi) - \hat{\mathcal{R}}_{\textit{GRPO}}(\pi) \right| \geq 2\mathfrak{R}_n(\mathcal{H}) + \sqrt{\frac{2 \log(2/\delta_1)}{n}} \right] \leq \delta_1 \tag{68}$$

*Substituting the Rademacher complexity bound:*

$$\mathbb{P}\left[ \sup_{\pi \in \mathcal{H}} \left| \mathcal{R}_{\textit{GRPO}}(\pi) - \hat{\mathcal{R}}_{\textit{GRPO}}(\pi) \right| \geq 2\sqrt{\frac{2d \log(2en/d)}{n}} + \sqrt{\frac{2 \log(2/\delta_1)}{n}} \right] \leq \delta_1 \tag{69}$$

*Using the inequality $\sqrt{a} + \sqrt{b} \leq \sqrt{2(a+b)}$ for $a, b \geq 0$:*

$$\mathbb{P}\left[\sup_{\pi \in \mathcal{H}} \left| \mathcal{R}_{GRPO}(\pi) - \hat{\mathcal{R}}_{GRPO}(\pi) \right| \geq \sqrt{\frac{8d \log(2en/d) + 8 \log(2/\delta_1)}{n}}\right] \leq \delta_1 \tag{70}$$

*Each preference reward is bounded:*

$$|R_i^{pref}| = \left| \sum_{t=2}^{|\tau_i|} \sigma(P_{\hat{\omega}_n}(\cdot)) \right| \leq \sum_{t=2}^{|\tau_i|} 1 = |\tau_i| - 1 \leq T_{\max} \tag{71}$$

*Since the preference model output is bounded by $|P_{\hat{\omega}_n}(\cdot)| \leq B$, and $\sigma(z) \in [0,1]$, we have:*

$$|R_i^{pref}| \leq BT_{\max} \tag{72}$$

*The preference-augmented loss function is:*

$$\ell_{pref}(\pi, q, \{\tau_i\}) = \frac{1}{G} \sum_{i=1}^{G} \frac{\lambda R_i^{pref}}{|\tau_i|} \sum_{t=1}^{|\tau_i|} \log \pi(a_{i,t}|s_{i,t}) \tag{73}$$

*Since $|\log \pi(a|s)| \leq \log(1/\pi_{\min}) \leq L_\pi$ for some constant $L_\pi$, the preference loss is bounded by:*

$$|\ell_{pref}(\pi, q, \{\tau_i\})| \leq \frac{1}{G} \sum_{i=1}^{G} \frac{\lambda BT_{\max}}{|\tau_i|} \cdot |\tau_i| \cdot L_\pi = \lambda BT_{\max}L_\pi \tag{74}$$

*Applying Hoeffding's inequality to the bounded preference rewards:*

$$\mathbb{P}\left[\left| \mathbb{E}[\ell_{pref}] - \hat{\mathbb{E}}[\ell_{pref}] \right| \geq t\right] \leq 2 \exp\left(-\frac{2nt^2}{(\lambda BT_{\max}L_\pi)^2}\right) \tag{75}$$

*Setting the right-hand side equal to $\delta_2$ and solving for $t$:*

$$t = \lambda BT_{\max}L_\pi \sqrt{\frac{\log(2/\delta_2)}{2n}} \tag{76}$$

*Absorbing $L_\pi$ into the bound and using a looser but cleaner bound:*

$$\mathbb{P}\left[\left| \mathbb{E}[R^{pref}] - \hat{\mathbb{E}}[R^{pref}] \right| \geq \lambda BT_{\max}\sqrt{\frac{2 \log(2/\delta_2)}{n}}\right] \leq \delta_2 \tag{77}$$

*From Theorems A.1 and A.2, the preference model error contributes an additional term. The preference model error is bounded by:*

$$\epsilon_{pref} = \|P_{\hat{\omega}_n} - P_{\omega^*}\|_\infty \leq \sqrt{\frac{2d_P \log(2en/d_P) + 2 \log(2/\delta_3)}{n}} \tag{78}$$

*This error propagates to the policy objective with the bound from Theorem A.2:*

$$|\mathbb{E}[J_{WS\text{-}GRPO}(\theta)] - \mathbb{E}[J^*(\theta)]| \leq \frac{\lambda T_{\max}}{4} \epsilon_{pref} \tag{79}$$

*Substituting the preference model error bound:*

$$|\mathbb{E}[J_{WS\text{-}GRPO}(\theta)] - \mathbb{E}[J^*(\theta)]| \leq \frac{\lambda T_{\max}}{4} \sqrt{\frac{2d_P \log(2en/d_P) + 2 \log(2/\delta_3)}{n}} \tag{80}$$

*This gives us:*

$$\mathbb{P}\left[\left|\mathbb{E}[J_{\textit{WS-GRPO}}(\theta)] - \mathbb{E}[J^*(\theta)]\right| \geq \frac{\lambda T_{\max}}{4}\sqrt{\frac{2d_P \log(2en/d_P) + 2\log(2/\delta_3)}{n}}\right] \leq \delta_3 \tag{81}$$

*Now we combine all error sources using the union bound. Setting $\delta_1 = \delta_2 = \delta_3 = \delta/3$ and applying the union bound, with probability at least $1 - \delta$:*

$$\mathcal{R}(\pi_\theta) - \hat{\mathcal{R}}(\pi_\theta) \leq |\mathcal{R}_{\textit{GRPO}}(\pi_\theta) - \hat{\mathcal{R}}_{\textit{GRPO}}(\pi_\theta)| \tag{82}$$
$$+ |\mathbb{E}[\ell_{\textit{pref}}] - \hat{\mathbb{E}}[\ell_{\textit{pref}}]| \tag{83}$$
$$+ |\mathbb{E}[J_{\textit{WS-GRPO}}(\theta)] - \mathbb{E}[J^*(\theta)]| \tag{84}$$

*Substituting the individual bounds:*

$$\mathcal{R}(\pi_\theta) - \hat{\mathcal{R}}(\pi_\theta) \leq \sqrt{\frac{8d\log(2en/d) + 8\log(6/\delta)}{n}} \tag{85}$$
$$+ \lambda B T_{\max}\sqrt{\frac{2\log(6/\delta)}{n}} \tag{86}$$
$$+ \frac{\lambda T_{\max}}{4}\sqrt{\frac{2d_P \log(2en/d_P) + 2\log(6/\delta)}{n}} \tag{87}$$

*To obtain a more compact form, we combine these three terms. Let $d_{\max} = \max(d, d_P)$ and observe that all terms have the same $O(\sqrt{\log n/n})$ rate. Using the inequality $\sqrt{a} + \sqrt{b} + \sqrt{c} \leq \sqrt{3(a+b+c)}$ and factoring out common terms:*

$$\mathcal{R}(\pi_\theta) - \hat{\mathcal{R}}(\pi_\theta) \leq \sqrt{\frac{8d\log(2en/d) + 8\log(12/\delta)}{n}} \tag{88}$$
$$+ \lambda B T_{\max}\sqrt{\frac{2\log(12/\delta)}{n}} \tag{89}$$
$$+ \frac{\lambda T_{\max}}{4}\sqrt{\frac{2d_P \log(2en/d_P) + 2\log(12/\delta)}{n}} \tag{90}$$
$$\leq \sqrt{\frac{C_1 d_{\max} \log(en/d_{\max}) + C_2 \lambda^2 (BT_{\max})^2 + C_3 \log(1/\delta)}{n}} \tag{91}$$

*where $C_1, C_2, C_3 > 0$ are universal constants that absorb the numerical factors. This compact form shows that the generalization error scales as:*

$$\mathcal{R}(\pi_\theta) - \hat{\mathcal{R}}(\pi_\theta) = \tilde{O}\left(\sqrt{\frac{d_{\max} + \lambda^2 (BT_{\max})^2}{n}}\right) \tag{92}$$

*where $\tilde{O}$ hides logarithmic factors in $n$ and $\delta$. This demonstrates that WS-GRPO maintains the standard statistical learning rate while the preference-specific terms contribute additively to the complexity, controlled by the mixing weight $\lambda$ and model capacities.*

*Table 3.* Training/Validation/Testing Split for datasets.

| Dataset | Training | Validation | Testing |
|---|---|---|---|
| ARC | 1813 | 129 | 648 |
| CommonsenseQA | 2000 | 200 | 2740 |
| Deepmath | 2000 | 200 | 2883 |
| GSM8K | 2000 | 200 | 2198 |

## Prompt 1: AI2-ARC Scientific Reasoning

**System Prompt:**
A conversation between User and Assistant. The User asks a question, and the Assistant solves it. The Assistant first thinks about the reasoning process in the mind and then provides the User with the answer. The reasoning process is enclosed within `<think> </think>` and answer is enclosed within `<answer> </answer>` tags, respectively, i.e., `<think>` reasoning process here `</think> <answer>` answer here`</answer>`. `<answer>` must contain only the letter of your choice (A, B, C, D).

**User Prompt:**
```
<Question>
<Options>
```

## Prompt 2: CommonsenseQA Reasoning

**System Prompt:**
A conversation between User and Assistant. The User asks a question, and the Assistant solves it. The Assistant first thinks about the reasoning process in the mind and then provides the User with the answer. The reasoning process is enclosed within `<think> </think>` and answer is enclosed within `<answer> </answer>` tags, respectively, i.e., `<think>` reasoning process here `</think> <answer>` answer here`</answer>`. `<answer>` must contain only the letter of your choice (A, B, C, D, or E).

**User Prompt:**
```
<Question>
<Options>
```

## Prompt 3: GSM8K and DeepMath Mathematical Reasoning

*GSM8K and DeepMath share the same system prompt since both target free-form mathematical reasoning with numeric answers.*

**System Prompt:**
A conversation between User and Assistant. The User asks a question, and the Assistant solves it. The Assistant first thinks about the reasoning process in the mind and then provides the User with the answer. The reasoning process is enclosed within `<think> </think>` and answer is enclosed within `<answer> </answer>` tags, respectively, i.e., `<think>` reasoning process here `</think> <answer>` answer here`</answer>`. `<answer>` must be numeric only (no words, no LaTeX, no commas, no units, no explanation, no trailing period).

**User Prompt:**
```
<Question>
```

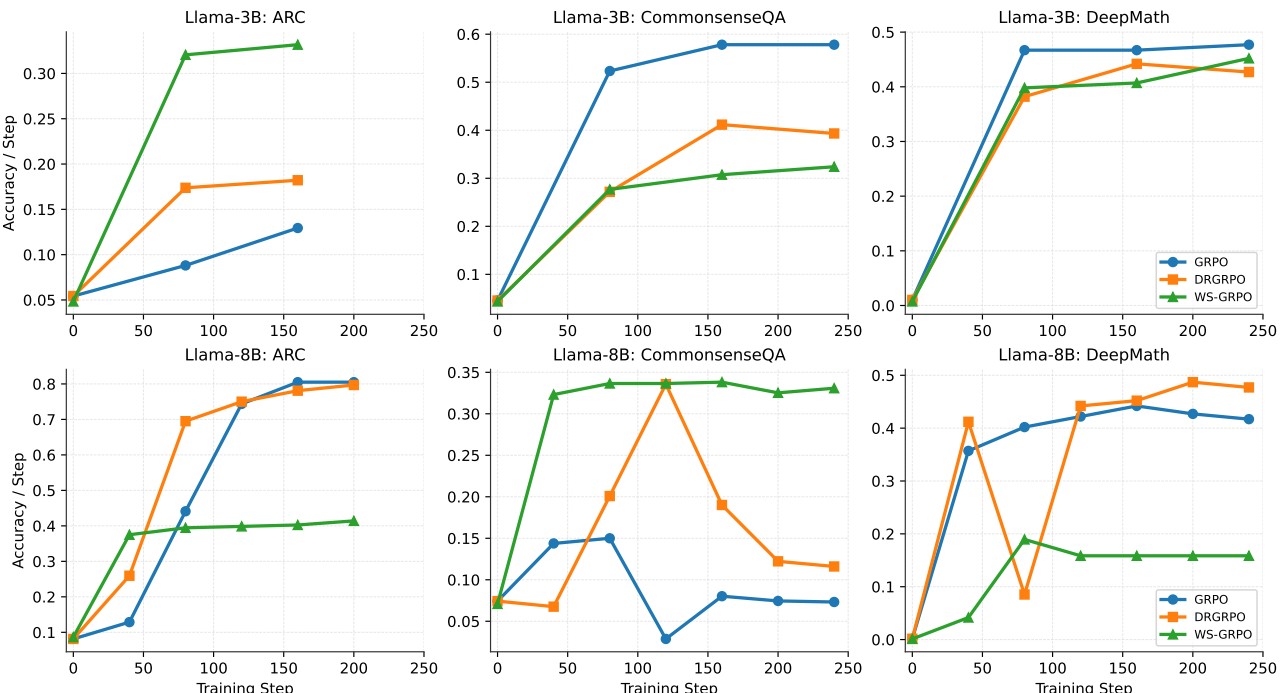

*Figure 5.* Validation step-efficiency (Pass@1 / average reasoning steps) across training for Llama models. Higher values indicate greater accuracy per reasoning step.

## A.2. Dataset Details

## A.3. Comparison with PACR

To address the concern that WS-GRPO's efficiency gains may reduce to general reward shaping, we compare against PACR (Yoon et al., 2025), which provides reward shaping via ground-truth confidence growth. Table 5 reports results on all four datasets with Qwen3-4B-Instruct.

PACR matches or slightly exceeds GRPO accuracy on ARC, GSM8K, and CommonsenseQA (PACR is within 0.2% on ARC and CSQA, and +2.8% on GSM8K relative to GRPO), but it provides no meaningful efficiency improvement: response lengths and step counts remain comparable to or exceed GRPO across all four datasets, including a substantial length increase on DeepMath (392.78 vs. 188.35 tokens). In contrast, WS-GRPO reduces eval length by 87.5% on ARC (20.23 vs. 161.94 tokens) and steps by 75.1% (2.00 vs. 8.04), while retaining 95.3% of the best baseline accuracy. On GSM8K, WS-GRPO achieves 0.917 Pass@1 with 114.53 tokens, compared to PACR's 0.934 at 155.23 tokens. As a unified metric, WS-GRPO achieves Pass@1/Steps ratios of 0.443 (ARC), 0.445 (GSM8K), 0.356 (CSQA), and 0.266 (DeepMath), compared to PACR's 0.096, 0.136, 0.190, and 0.053 respectively—a 1.9–5.0× advantage in step-efficiency. These results confirm that WS-GRPO's contribution is the learned preference signal over consecutive prefixes, not reducible to general reward shaping.

## A.4. Accuracy–Efficiency Pareto Analysis

To complement the step-efficiency metric in the main text, we visualize the accuracy–cost trade-off directly via Pareto-style scatter plots. Figure 6 plots test Pass@1 against average reasoning steps (a) and response length in tokens (b) across all four models and four datasets. Each point is one (model, dataset) configuration; gray connectors link the three methods evaluated on the same configuration.

WS-GRPO consistently occupies the upper-left (Pareto-optimal) region—high accuracy at low cost. The green band at $s^* = 2$ steps marks WS-GRPO's target operating point, and nearly all WS-GRPO configurations cluster there. GRPO and DRGRPO, by contrast, span 1–15 steps with no systematic efficiency advantage. On ARC, where baselines produce verbose multi-step outputs, WS-GRPO achieves 85.7–92.9% token reductions on Qwen2.5-7B-Instruct, Qwen3-4B-Instruct, and

*Table 4.* **Main results (accuracy + efficiency) for GRPO vs DRGRPO vs WS-GRPO.** We report **test-set** Pass@1 accuracy (↑) and **test-set** average reasoning steps (↓). Efficiency is additionally reported as the **mean completion length** in tokens (↓) measured during evaluation at the **final validation checkpoint**. **Improve** compares WS-GRPO against the best baseline among {GRPO, DRGRPO} for each metric (max for Pass@1; min for length/steps). Green cells indicate improvements in cost metrics (lower steps/length).

| Dataset | Metric | Llama-3.2-3B-Instruct | | | | Llama-3.1-8B-Instruct | | | |
|---|---|---|---|---|---|---|---|---|---|
| | | GRPO | DRGRPO | WS-GRPO | Improve | GRPO | DRGRPO | WS-GRPO | Improve |
| ARC | Pass@1 ↑ | 0.794 | 0.782 | **0.748** | $\Delta = -0.046$ | 0.816 | 0.841 | **0.824** | $\Delta = -0.017$ |
| | Eval length (tok.) ↓ | 137.20 | 97.91 | **14.00** | ↓ 85.7% | 13.05 | 8.00 | **20.00** | ↑ 150% |
| | Avg steps ↓ | 5.07 | 4.22 | **2.04** | ↓ 51.7% | 1.00 | 1.00 | **2.00** | ↑ 100% |
| CommonsenseQA | Pass@1 ↑ | 0.714 | 0.717 | **0.688** | $\Delta = -0.029$ | 0.737 | 0.749 | **0.721** | $\Delta = -0.028$ |
| | Eval length (tok.) ↓ | 38.20 | 51.60 | **37.49** | ↓ 1.9% | 165.03 | 441.09 | **38.44** | ↓ 76.7% |
| | Avg steps ↓ | 1.17 | 1.80 | **2.01** | ↑ 71.8% | 9.36 | 5.81 | **2.11** | ↓ 63.7% |
| DeepMath | Pass@1 ↑ | 0.455 | 0.460 | **0.464** | $\Delta = +0.004$ | 0.464 | 0.463 | **0.372** | $\Delta = -0.092$ |
| | Eval length (tok.) ↓ | 9.02 | 8.14 | **12.66** | ↑ 55.5% | 9.09 | 512.00 | **41.49** | ↑ 356% |
| | Avg steps ↓ | 1.00 | 1.01 | **1.00** | ↓ 0% | 1.00 | 1.00 | **2.00** | ↑ 100% |
| GSM8K | Pass@1 ↑ | 0.789 | 0.731 | **0.798** | $\Delta = +0.009$ | 0.853 | 0.837 | **0.845** | $\Delta = -0.008$ |
| | Eval length (tok.) ↓ | 172.79 | 216.02 | **153.75** | ↓ 11.0% | 164.43 | 512.00 | **193.51** | ↑ 17.7% |
| | Avg steps ↓ | 6.88 | 5.28 | **2.24** | ↓ 57.6% | 5.81 | 2.63 | **2.09** | ↓ 20.5% |

Llama-3.2-3B-Instruct while remaining within 2.5–4.6% accuracy of the best baseline; on Llama-3.1-8B-Instruct the ARC baseline is already concise (∼13 tokens, 1 step), so WS-GRPO maintains rather than reduces length. On GSM8K, length reductions are smaller (11.0–52.5% across the three over-generating models), reflecting the milder baseline redundancy on this task.

Figures 7 and 8 disaggregate the same trade-offs by model family, revealing that the efficiency gains are consistent across architectures and scales. Figure 9 quantifies the trade-off as paired bar charts: accuracy retention (WS-GRPO Pass@1 divided by the best of GRPO/DRGRPO) and step reduction (percentage decrease relative to the mean of GRPO and DRGRPO steps). Across all 16 configurations, 11 retain ≥95% accuracy relative to the best baseline, and 10 achieve ≥50% step reduction relative to the mean baseline.

## A.5. Extended Comparison to Closely Related Methods

Several recent methods study intermediate credit assignment or efficient reasoning under GRPO-style training, but they differ from WS-GRPO in the source of supervision, the target setting, and whether they modify the rollout pipeline. S-GRPO targets efficient reasoning through early exit by constructing serial groups from different exit positions along a single reasoning path and assigning rule-based decaying rewards to correct early exits (Dai et al., 2026). Its strength is an explicit early-exit objective, but the signal depends on a specialized serial-group rollout procedure and a hand-designed decay schedule. WS-GRPO instead learns a content-aware preference model from correct and incorrect outcomes and uses it to produce prefix-level pseudo-rewards without imposing a fixed decay rule over exit positions.

Segment Policy Optimization (SPO) addresses credit assignment at the segment level, sitting between token-level and trajectory-level reinforcement learning (Guo et al., 2026). Its strength is localized Monte Carlo segment advantage estimation, which can provide finer credit assignment than standard trajectory-level GRPO. However, SPO is not primarily an efficiency-control method and its segment-level Monte Carlo estimation adds sampling cost during training. WS-GRPO instead evaluates consecutive prefixes using a learned correctness-aware preference signal derived from outcome-only supervision.

SPA-RL studies stepwise progress attribution for language-model agents that interact with external environments (Wang et al., 2025a). Its progress estimator is useful when actions have environment-grounded executability and task-completion signals. This setting is materially different from pure text reasoning: WS-GRPO does not assume external environment interaction or executable actions, and instead converts final-answer correctness into weak prefix-level guidance over chain-of-thought trajectories.

PACR uses confidence-growth based reward shaping for LLM reasoning (Yoon et al., 2025). It is a relevant reward-shaping baseline because it also introduces intermediate signals, but its supervision relies on confidence progression rather than a

*Table 5.* **PACR comparison on Qwen3-4B-Instruct.** PACR matches or slightly exceeds GRPO accuracy on ARC, CSQA, and GSM8K but provides no efficiency gain. WS-GRPO achieves 1.9–5.0× higher step-efficiency (Pass@1/Steps).

| Dataset | Metric | GRPO | DRGRPO | WS-GRPO | PACR |
|---|---|---|---|---|---|
| ARC | Pass@1 ↑ | 0.930 | 0.926 | 0.886 | **0.929** |
| | Length (tok.) ↓ | 161.94 | 268.47 | **20.23** | 205.39 |
| | Avg steps ↓ | 8.04 | 12.34 | **2.00** | 9.71 |
| CSQA | Pass@1 ↑ | 0.774 | **0.797** | 0.768 | 0.776 |
| | Length (tok.) ↓ | **14.04** | 220.59 | 42.64 | 87.39 |
| | Avg steps ↓ | **1.00** | 10.83 | 2.16 | 4.08 |
| GSM8K | Pass@1 ↑ | 0.906 | 0.933 | 0.917 | **0.934** |
| | Length (tok.) ↓ | 161.17 | 146.52 | **114.53** | 155.23 |
| | Avg steps ↓ | 7.79 | 1.12 | **2.06** | 6.86 |
| DeepMath | Pass@1 ↑ | **0.620** | 0.612 | 0.532 | 0.421 |
| | Length (tok.) ↓ | 188.35 | 185.17 | **16.29** | 392.78 |
| | Avg steps ↓ | 7.09 | 7.29 | **2.00** | 7.98 |

*Table 6.* Qualitative comparison with closely related intermediate-credit-assignment and efficient-reasoning methods.

| Method | Targets Rollout Efficiency | Learned Signal | Outcome-Only Supervision | Main Limitation / Difference |
|---|---|---|---|---|
| S-GRPO (Dai et al., 2026) | Yes | No | Yes | Rule-based exponential decay tied to exit order; requires serial-group rollout construction. |
| SPO (Guo et al., 2026) | No | No | Yes | Segment-level Monte Carlo estimation adds training-time sampling cost. |
| SPA-RL (Wang et al., 2025a) | No | Yes | No | Designed for agent-environment interaction and executable actions. |
| PACR (Yoon et al., 2025) | Partially | No | Yes | Uses confidence-growth reward shaping rather than learned prefix preferences. |
| WS-GRPO | Yes | Yes | Yes | In-domain preference model learned from final-answer correctness. |

learned preference model over consecutive prefixes. WS-GRPO is designed to identify whether a continuation improves correctness relative to the preceding prefix, which directly targets redundant continuation and over-generation.

Closer in spirit to WS-GRPO, Fanconi et al. (2026) learn reasoning reward models from expert demonstrations via inverse reinforcement learning, deriving a dense reward signal from high-quality reference trajectories. The supervision sources differ in a way that matters for deployment: their method assumes access to expert demonstrations, whereas WS-GRPO requires only final-answer correctness on policy-generated rollouts and constructs prefix-level guidance entirely from outcome labels. The two directions are complementary — expert-demonstration IRL can shape the reward when trusted trajectories exist, while WS-GRPO is applicable in settings where only verifiable outcomes are available.

### A.6. Length Penalty

To encourage reasoning trajectories with appropriate length, we apply a length penalty to the stepwise reward:

$$\ell(n) = \begin{cases} -\alpha(n_{\min} - n) & \text{if } n < n_{\min} \\ 0 & \text{if } n_{\min} \leq n \leq n_{\max} \\ -\alpha(n - n_{\max}) & \text{if } n > n_{\max} \end{cases} \tag{93}$$

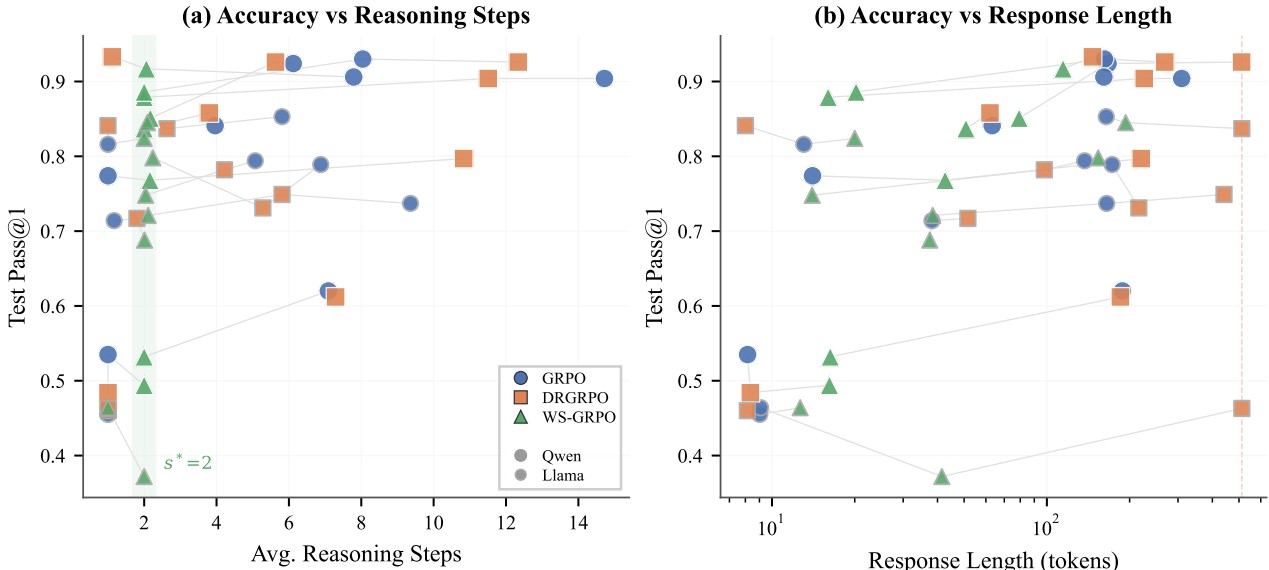

*Figure 6.* **Accuracy vs. reasoning cost (all models and datasets).** Each point is one (model, dataset) configuration; gray connectors link the three methods on the same configuration. **(a)** Pass@1 vs. average reasoning steps; the green band marks the $s^* = 2$ target. **(b)** Pass@1 vs. response length (tokens, log scale). WS-GRPO (green triangles) consistently clusters in the upper-left Pareto-optimal region, achieving competitive accuracy at substantially lower cost.

where $n$ is the number of reasoning steps, $n_{\min} = 3$, $n_{\max} = 6$, and $\alpha = 0.1$. The adjusted stepwise reward is computed as:

$$r_{\text{step}} = \frac{\bar{p} + \ell(n)}{n} \tag{94}$$

where $\bar{p}$ is the mean preference probability across steps. This normalization ensures fair comparison across trajectories of different lengths.

### A.7. Training Hyperparameters

**Phase I - Preference Learning:** We use Qwen2.5-3B-Instruct to generate 4 reasoning trajectories per question, yielding an average of 85,425 preference pairs per dataset (ranging from 60K to 103K). We apply a 90%/10% train-validation split. The preference model uses a frozen FLAN-T5 encoder followed by a lightweight MLP classifier with hidden dimension 512, with batch size 32 and learning rate $5 \times 10^{-5}$.

**Phase II - Policy Optimization:** We use $G = 8$ rollouts per prompt with learning rate $\eta = 1 \times 10^{-5}$. The mixing weight is set to $\lambda = 0.1$ to balance preference rewards with outcome correctness. Length penalty coefficient $\alpha = 0.1$ encourages trajectories between 3 and 6 reasoning steps (see Section A.6 for details). Step-wise rewards are normalized by trajectory length to prevent bias toward longer sequences.

### A.8. Ability to anticipate final correctness for preference model

We further designed a fine-grained metric to assess the model's ability to anticipate final correctness from partial trajectories. For each pair of correct and incorrect trajectories of length $n$, we evaluate all prefixes of length $2, 3, \ldots, n$. A correctly ordered prefix pair receives a weight inversely proportional to its step index, emphasizing early correct predictions. The cumulative score is normalized by the maximum achievable value. Figure 10 shows that this metric increases with trajectory length, confirming that the model not only distinguishes full trajectories but also reliably ranks partial reasoning paths. The upward trend suggests that longer trajectories provide more discriminative signal, enabling more accurate stepwise reward assignment during policy optimization.

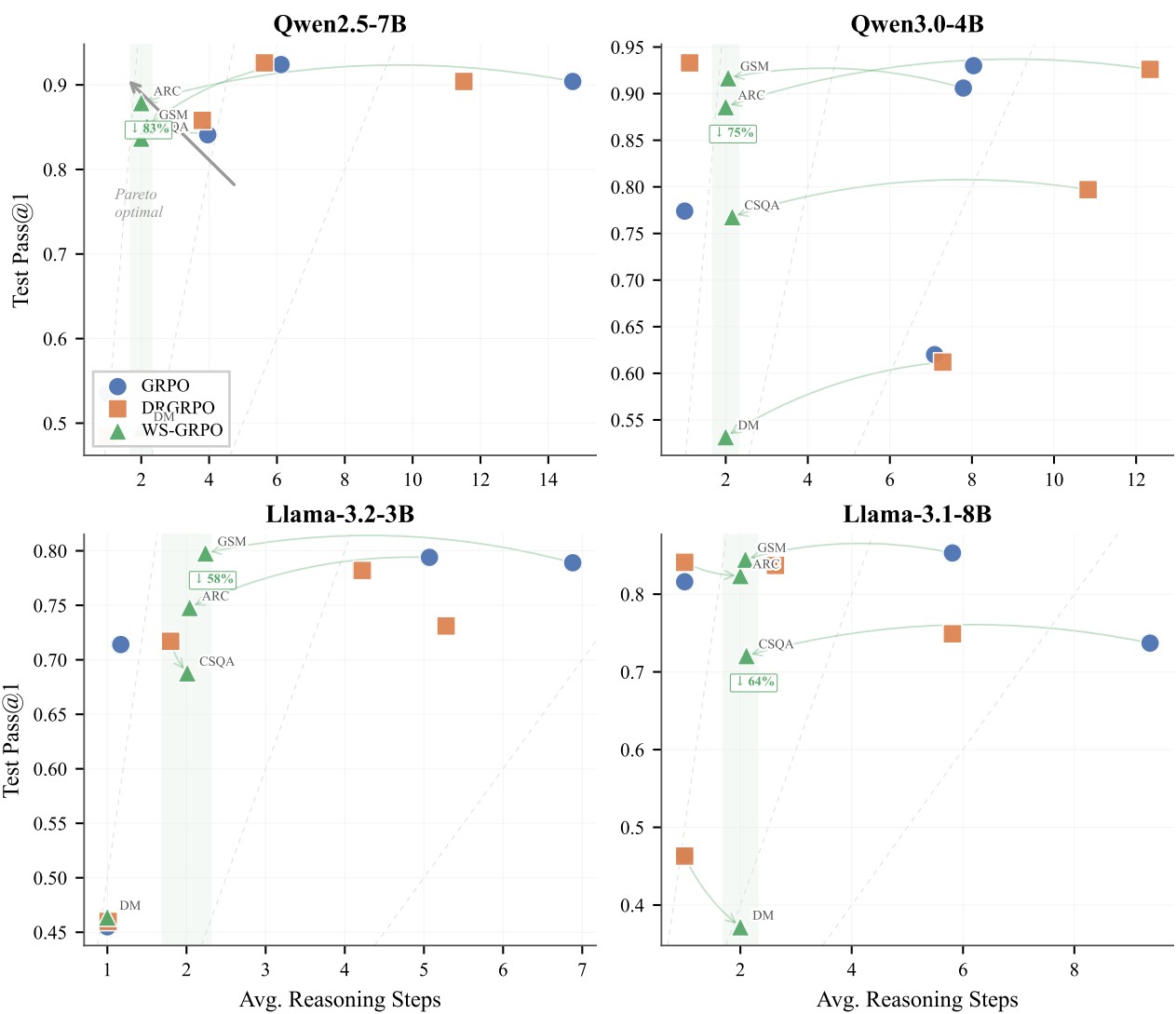

*Figure 7.* **Per-model Pareto: accuracy vs. reasoning steps.** Each panel shows one model family; arrows connect each baseline to WS-GRPO on the same dataset. Dataset labels (ARC, CSQA, DM, GSM) appear next to WS-GRPO markers. Iso-efficiency lines (Pass@1/steps) are shown as dashed gray curves; higher is better. WS-GRPO achieves competitive accuracy at ≈2 steps across nearly all configurations.

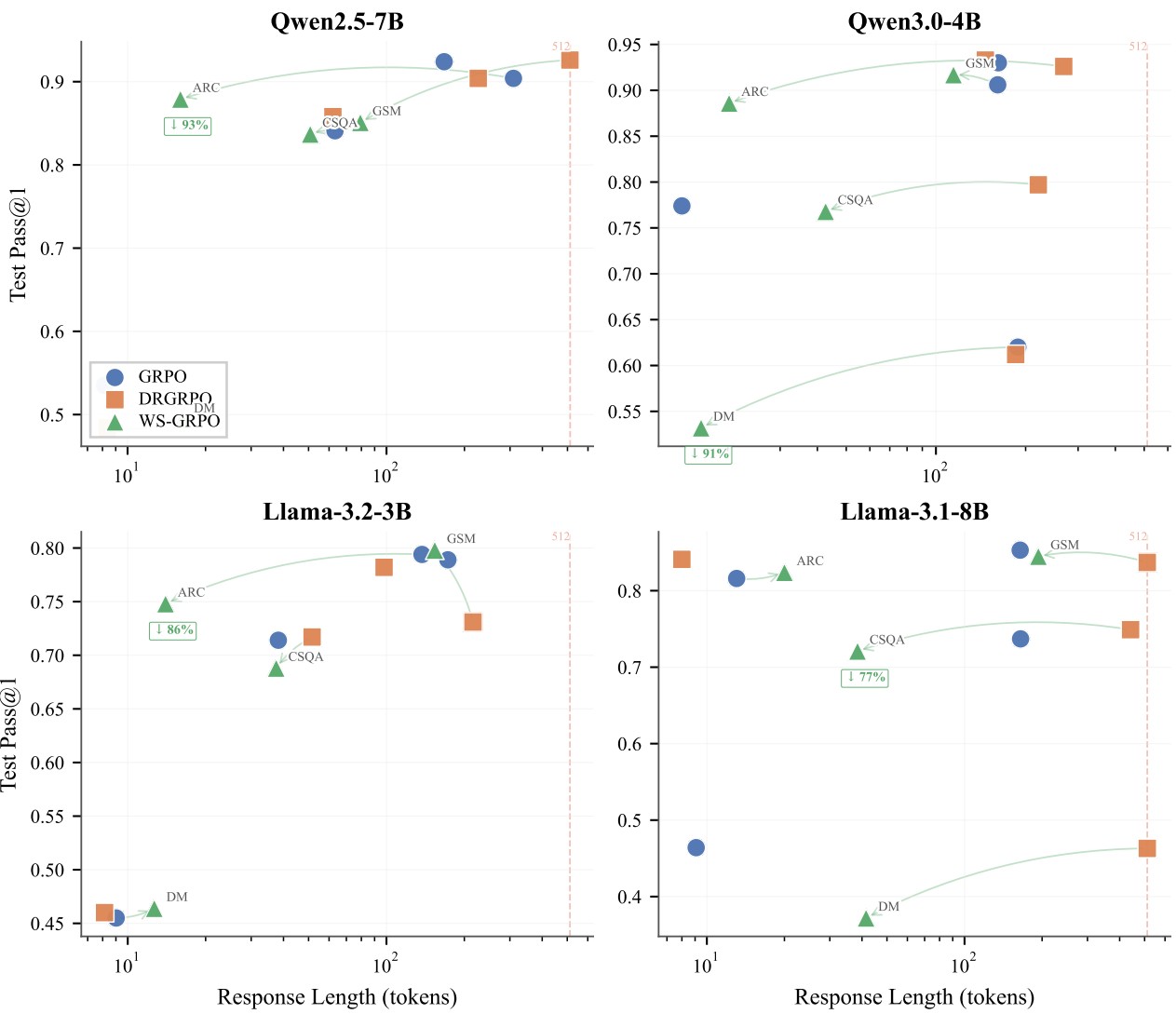

*Figure 8.* **Per-model Pareto: accuracy vs. response length** (tokens, log scale). Arrows connect each baseline to WS-GRPO on the same dataset. The dashed red line at 512 tokens marks the maximum-length clipping boundary observed for some DRGRPO configurations. WS-GRPO achieves order-of-magnitude length reductions on ARC across all model families.

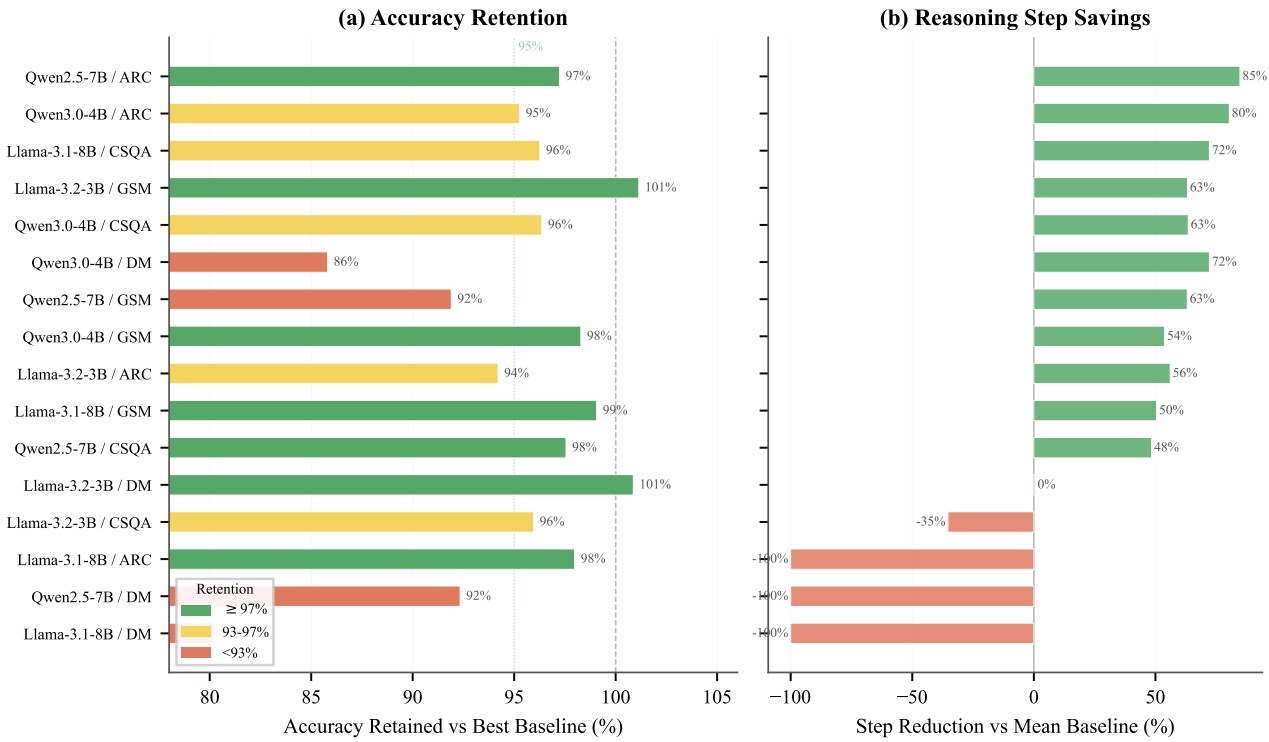

*Figure 9.* **Accuracy retention and step reduction across all 16 (model, dataset) configurations.** (a) Accuracy retained relative to the best baseline (higher is better). (b) Reasoning step savings relative to the mean baseline (higher is better). Green bars indicate $\geq 97\%$ retention; yellow 93–97%; red <93%.

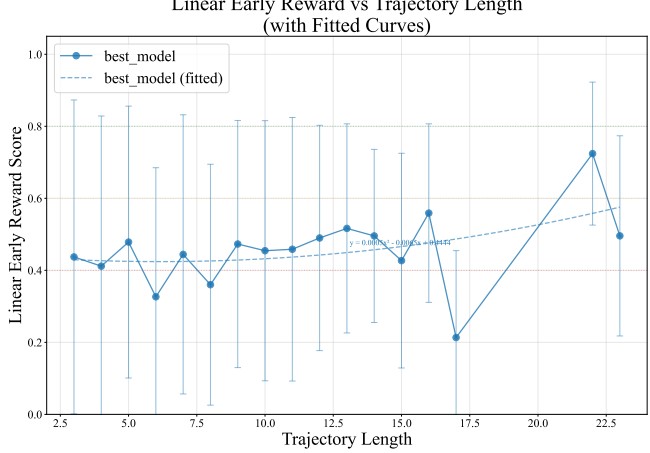

*Figure 10.* Preference Model Effectiveness

