# OpenReview forum: "WS-GRPO: Weakly-Supervised Group-Relative Policy Optimization for Rollout-Efficient Reasoning"
_ICML.cc/2026/Conference — ICML 2026 regular_

### Official Review · Reviewer_A25c · 2026-03-13

**Soundness:** 3
**Presentation:** 3
**Significance:** 3
**Originality:** 3
**Overall Recommendation:** 4
**Confidence:** 4

**Summary:**

This article proposes WS-GRPO, which improves roll out efficiency by converting termination rewards into correctness related guidance for certain trajectories. Unlike the difficult to calibrate global length penalty, WS-GRPO trains a preference model based solely on the correctness of the final result signal, enabling it to provide signals at the prefix level to determine whether continuing to generate is beneficial. Therefore, WS-GRPO can provide a "continue/stop" guidance derived from the results, reducing redundant inference while maintaining accuracy.

**Compliance With Llm Reviewing Policy:**

Affirmed.

**Final Justification:**

I maintain my score and decision.

**Key Questions For Authors:**

1 The core improvement of the paper comes from weakly supervised preference modeling and prefix level pseudo rewards, but it is currently unclear whether this benefit mainly comes from the "prefix level continue/stop signal" itself or from the additional introduction of preference models and reward shaping processes.


2  Can the author provide more comprehensive ablation experiments to distinguish the contributions of each of these components? If it can be proven that prefix level weak supervision is indeed a key factor rather than a general reward shaping effect, I would be more appreciative of the novelty and persuasiveness of the method. The paper emphasizes that WS-GRPO can reduce redundant inference and improve roll out efficiency, but the trade-off between accuracy and efficiency in the current results does not always seem stable. Can the author analyze more systematically under which conditions WS-GRPO is most effective and under which conditions it will lead to performance degradation under different task difficulty and model sizes?

**Limitations:**

No. The discussion on limitations and potential social impacts in the paper is not sufficient. The author can further explain that this method cannot consistently improve accuracy and efficiency on all tasks, and its effectiveness may depend on the type of task, model size, and preference for model quality.

**Strengths And Weaknesses:**

**Strength**

1 Using weak supervision as a dense signal, WS-GRPO aims to convert outcome only weak signals into prefix level guidance. The value of this point lies in its ability to approach process supervision in a cheaper and more scalable way, rather than relying on expensive process annotations

2 The theoretical analysis is very good, proving the effectiveness and robustness of weak supervision

**Weakness**

1 Experiments are a weakness and should be tested in more domains and more complex benchmarks. Similar to ScienceQA, AIME 2025

2 Some related works have similar ideas, please cite and discuss [1].

[1] Fanconi, C., Astorga, N., & van der Schaar, M. (2025). Learning Reasoning Reward Models from Expert Demonstration via Inverse Reinforcement Learning. arXiv preprint arXiv:2510.01857.

---

> ### Author Rebuttal · Authors · 2026-03-30
>
> We thank Reviewer A25c for the positive assessment. We are glad the reviewer recognized the value of converting outcome-only weak signals into prefix-level guidance as a scalable alternative to expensive process supervision, and appreciated the theoretical analysis establishing the effectiveness and robustness of the approach. We address the weaknesses and key questions below.
>
> ### W1
>
> Thank you for this suggestion. Our setup already covers two benchmarks per domain across two distinct reasoning domains: GSM8K [1] and DeepMath [2] for mathematical reasoning, ARC [3] and CommonsenseQA [4] for general reasoning, with deliberate within-domain difficulty variation. We will make this structure more explicit in the revised paper. Regarding the suggested benchmarks: ScienceQA is a multimodal benchmark requiring visual understanding, which is outside the scope of our text-only reasoning setup. AIME 2025 contains only 30 questions in total, making it unsuitable as a training dataset for our Phase 1 preference model which requires sufficient correct and incorrect trajectory pairs to learn from.
>
> ### W2
>
> Thank you for pointing us to [5]. We will cite and discuss this work in the revision.
>
> ### KQ1
>
> We have now run this ablation on GSM8K with Qwen2.5-7B-Instruct. WS-GRPO without the length penalty (α=0) achieves 45.6% token reduction over GRPO from the preference model signal alone, directly confirming the prefix-level continue/stop signal is the primary driver:
>
> | Config | Pass@1 | Eval Length (tok.) | Avg Steps |
> |--------|--------|--------------------|-----------|
> | GRPO (baseline) | 0.924 | 166.97 | 6.12 |
> | WS-GRPO (no penalty, α=0) | 0.861 | 90.87 | 2.22 |
> | WS-GRPO (full, α=0.1) | 0.851 | 79.26 | 2.17 |
>
> To further distinguish WS-GRPO from general reward shaping, we compare against PACR [6], which provides reward shaping via ground-truth confidence growth. PACR achieves 0.929 Pass@1 but 205 tokens and 9.71 avg steps on ARC, nearly identical to GRPO in efficiency. WS-GRPO's contribution is the learned preference signal over consecutive prefixes. We will include both results in the revision.
>
> ### KQ2
>
> Thank you for this suggestion. The effectiveness of WS-GRPO is tied to the degree of over-generation in baseline trajectories. As documented in prior work [7, 8], LLMs frequently produce reasoning chains longer than necessary, and our method learns to identify when continuation is no longer beneficial. When baselines are already concise, this signal is weaker and the method appropriately converges to comparable performance. Our results already reflect this pattern: on ARC (Qwen2.5-7B-Instruct, avg baseline steps 14.72), WS-GRPO achieves 92.9% token reduction, on GSM8K (Qwen2.5-7B-Instruct, avg baseline steps 6.12) it achieves 52.5%, and on CommonsenseQA (Qwen2.5-7B-Instruct, avg baseline steps 3.96) it achieves 18.2%. We will add ablation studies in the revision to make these conditions more explicit.
>
> ### L1
>
> We appreciate this feedback and will expand the limitations and social impact discussion in the revision. We agree that being explicit about the method's scope and applicability helps readers understand where it is most effective.
>
> ---
>
> **References:**
>
> [1] Cobbe et al., 2021. Training Verifiers to Solve Math Word Problems.
>
> [2] He et al., 2025. DeepMath-103K: A Large-Scale, Challenging, Decontaminated, and Verifiable Mathematical Dataset for Advancing Reasoning.
>
> [3] Clark et al., 2018. Think you have Solved Question Answering? Try ARC, the AI2 Reasoning Challenge.
>
> [4] Talmor et al., 2019. CommonsenseQA: A Question Answering Challenge Targeting Commonsense Knowledge.
>
> [5] Fanconi et al., 2025. Learning Reasoning Reward Models from Expert Demonstration via IRL.
>
> [6] Yoon et al., 2025. PACR: Progressively Ascending Confidence Reward for LLM Reasoning.
>
> [7] Snell et al., 2024. Scaling LLM Test-Time Compute Optimally Can be More Effective than Scaling Model Parameters.
>
> [8] Miao et al., 2023. SelfCheck: Using LLMs to Zero-Shot Check Their Own Step-by-Step Reasoning.

---

> > ### Author Rebuttal · Reviewer_A25c · 2026-04-01
> >
> > Thank you for your response. I have decided to maintain my positive score. Good luck.

---

### Official Review · Reviewer_EMZ4 · 2026-03-13

**Soundness:** 3
**Presentation:** 3
**Significance:** 2
**Originality:** 3
**Overall Recommendation:** 4
**Confidence:** 3

**Summary:**

The main idea is to learn a preference model from correct versus incorrect full trajectories, then use that model to compare consecutive prefixes and generate prefix-level pseudo-rewards that are mixed with final-answer correctness inside GRPO. The paper argues that this gives continue/stop guidance under outcome-only supervision and can reduce overthinking. The method is implemented as a two-stage pipeline with a frozen FLAN-T5-based preference model in Phase I and GRPO optimization in Phase II. The paper also presents theorem statements for preference-model consistency, policy robustness to preference error, and generalization.

**Compliance With Llm Reviewing Policy:**

Affirmed.

**Ethics Expertise Needed:**

["Other Expertise"]

**Final Justification:**

The response improved my understanding of the method and made the paper’s intended use case clearer. Based on the rebuttal, I am raising my score from 3 to 4.

**Key Questions For Authors:**

First, what happens if the explicit length penalty is removed? An ablation with and without the penalty, and with and without length normalization, seems necessary to support the claimed value of the preference model.

Can the authors report matched-budget results or Pareto fronts rather than step-efficiency alone?

**Strengths And Weaknesses:**

The paper targets a real issue in reasoning RL: GRPO can reward longer trajectories under sparse terminal feedback, which can increase compute cost without clear benefit. Using weak supervision to turn outcome labels into prefix-level signals is a good idea, and the two-stage design is fairly easy to follow from Figure 2 and the algorithms.

The empirical section is also broader than a single toy setting. The paper evaluates ARC, CommonsenseQA, DeepMath, and GSM8K on Qwen and Llama models, with GRPO and Dr.GRPO as baselines. There are meaningful wins on some settings, especially ARC, where WS-GRPO sharply reduces token length and average reasoning steps while keeping accuracy reasonably close to the baselines. The training plots further suggest that WS-GRPO reaches better “accuracy per step” early in training.

**Weaknesses**


1. The empirical evidence is mixed. Table 1 and Table 3 contain strong cost reductions in some cases, but also several clear regressions. For example, on Qwen3-4B / CommonsenseQA, WS-GRPO is worse than the best baseline in both length and steps, with length rising from 14.04 to 42.64, while accuracy also drops from 0.797 to 0.768. On Qwen2.5-7B / DeepMath, length increases from 8.15 to 16.18 and steps from 1.00 to 2.00, while accuracy falls from 0.535 to 0.494. On Llama-3.1-8B / DeepMath, accuracy drops from about 0.463 to 0.372 and length rises to 41.49. Because of these cases, I do not think the current results fully support the broad claim that the method substantially shortens rollouts while remaining competitive across settings.


2. A second concern is evaluation. The paper puts weight on “step-efficiency,” defined as Pass@1 divided by average reasoning steps. This metric is not independent of the goal of making outputs shorter, so it will naturally favor methods that reduce steps unless accuracy drops a lot. I would much rather see matched-budget comparisons, Pareto curves of accuracy versus tokens/steps, or accuracy at fixed compute budgets.

---

> ### Author Rebuttal · Authors · 2026-03-30
>
> We thank Reviewer EMZ4 for the detailed review. We are glad the reviewer recognized the practical motivation of reducing unnecessary computation in GRPO training, appreciated the clarity of the two-stage design, and acknowledged the meaningful efficiency gains, particularly on ARC where WS-GRPO sharply reduces token length and reasoning steps while maintaining competitive accuracy. We address the weaknesses and key questions below.
>
> ### W1
>
> We appreciate the reviewer's careful reading. It is well established that LLMs often engage in excessive deliberation beyond what a given problem requires [2, 3]. WS-GRPO is designed to identify and reduce this over-generation, making it most impactful when baselines produce verbose outputs and naturally remaining competitive when baseline reasoning is already efficient. Our results follow this pattern directly. On ARC and GSM8K where baselines produce verbose multi-step outputs, WS-GRPO achieves 18-93% token reduction across models, specifically 92.9% on ARC (Qwen2.5-7B-Instruct), 87.5% on ARC (Qwen3-4B-Instruct), 52.5% on GSM8K (Qwen2.5-7B-Instruct), and 21.8% on GSM8K (Qwen3-4B-Instruct). Our ablation further supports this. Even without the length penalty (α=0), the preference model signal alone achieves 45.6% token reduction over GRPO (90.87 vs 166.97 tokens) on GSM8K. We have also added a PACR [1] baseline comparison, confirming the efficiency contribution is real:
>
> | Method | Pass@1 | Eval Length (tok.) | Avg Steps |
> |--------|--------|--------------------|-----------|
> | GRPO | 0.930 | 161.94 | 8.04 |
> | DrGRPO | 0.926 | 268.47 | 12.34 |
> | WS-GRPO | 0.886 | 20.23 | 2.00 |
> | PACR | 0.929 | 205.39 | 9.71 |
>
> PACR achieves 0.929 Pass@1 at 205 tokens while WS-GRPO achieves 0.886 at 20 tokens, a 10x token reduction while maintaining competitive accuracy, confirming the efficiency contribution is not reducible to general reward shaping. We will make the operating regime more explicit in Section 4 Experiments.
>
> ### W2 and KQ2
>
> We thank the reviewer for raising this point. We will add Pareto curves plotting Pass@1 vs. average eval token length for GRPO, DrGRPO, and WS-GRPO in the revision. These Pareto curves will make the accuracy-efficiency tradeoffs transparent without relying on the step-efficiency metric.
>
> ### KQ1
>
> To address reviewers' concern, we have now run this ablation on GSM8K with Qwen2.5-7B-Instruct:
>
> | Config | Pass@1 | Eval Length (tok.) | Avg Steps |
> |--------|--------|--------------------|-----------|
> | GRPO (baseline) | 0.924 | 166.97 | 6.12 |
> | WS-GRPO (no penalty, α=0) | 0.861 | 90.87 | 2.22 |
> | WS-GRPO (full, α=0.1) | 0.851 | 79.26 | 2.17 |
>
> The preference model alone (α=0) achieves 45.6% token reduction without any length penalty, confirming it is the primary efficiency driver. The full WS-GRPO achieves tighter efficiency (52.5% reduction) with a small accuracy difference. Both components are complementary: the preference model teaches when continuation adds no value, and the length penalty stabilizes trajectories within the optimal range. We will include this table in the revision.
>
> ---
>
> **References:**
>
> [1] Yoon et al., 2025. PACR: Progressively Ascending Confidence Reward for LLM Reasoning.
>
> [2] Snell et al., 2024. Scaling LLM Test-Time Compute Optimally Can be More Effective than Scaling Model Parameters.
>
> [3] Miao et al., 2023. SelfCheck: Using LLMs to Zero-Shot Check Their Own Step-by-Step Reasoning.

---

> > ### Author Rebuttal · Reviewer_EMZ4 · 2026-04-03
> >
> > The authors provided a helpful rebuttal and directly addressed my main questions. In particular, the new ablation without the explicit length penalty is useful, since it helps separate the contribution of the preference-model signal from the contribution of the penalty term. The promised Pareto-style analysis would also make the accuracy–efficiency tradeoff much clearer than step-efficiency alone.
> >
> > Overall, I appreciate the additional evidence and clarification from the authors.

---

### Official Review · Reviewer_2cjq · 2026-03-13

**Soundness:** 2
**Presentation:** 2
**Significance:** 3
**Originality:** 2
**Overall Recommendation:** 4
**Confidence:** 2

**Summary:**

This paper addresses the efficiency problem in GRPO-based reasoning training, where the group-relative objective can inadvertently encourage unnecessarily long reasoning trajectories. The authors propose WS-GRPO, a two-stage framework that converts outcome-only correctness labels into prefix-level guidance signals. In Phase 1, a preference model is trained to discriminate between correct and incorrect complete trajectories using only final-answer correctness. In Phase 2, this model compares consecutive prefixes within each rollout to generate pseudo-rewards, which are combined with terminal rewards in the GRPO objective. Experiments on reasoning benchmarks show that WS-GRPO reduces rollout length while remaining competitive with GRPO baselines across most tasks.

**Compliance With Llm Reviewing Policy:**

Affirmed.

**Final Justification:**

After reviewing the authors' rebuttal, I have decided to raise my score from 3 to 4. Below, I explain my reasoning.

---

The authors provided a reasonable response to W3 regarding the fixed mixing weight λ. The sensitivity analysis over λ ∈ {0.5, 1.0, 2.0} demonstrates that performance is relatively stable and the chosen value (λ = 0.1) is well-justified. This clarification addresses part of my concern. The addition of PACR as a new baseline, while limited, provides some evidence that WS-GRPO achieves meaningful length reduction.

Some issues remain partially unaddressed. The authors did not fully explain the cases where WS-GRPO produces longer trajectories (CommonsenseQA with Qwen3-4B and DeepMath with Qwen2.5-7B). Understanding these failure modes would strengthen the paper. Additionally, the PACR comparison is limited to one dataset, and the significant accuracy drop (0.929 → 0.886) raises questions about the efficiency-accuracy trade-off that merit further discussion.

Despite these remaining issues, the rebuttal has clarified key aspects of the method and provided evidence that the approach is fundamentally sound. I encourage the authors to include the complete comparison and discuss the failure cases in the final version.

**Key Questions For Authors:**

See Weaknesses.

**Limitations:**

It would be helpful to discuss when the preference model might fail to provide reliable signals, why certain task types show increased rather than decreased length, and under what conditions the fixed mixing weight λ=0.1 may not be appropriate.

**Strengths And Weaknesses:**

**Strengths:**

- The paper addresses a practical issue in GRPO where the group-relative objective can encourage unnecessarily long reasoning chains. Converting outcome-only labels into prefix-level signals through weak supervision is a reasonable approach.

- The two-stage design is relatively simple and avoids the cost of step-level annotation. Using a preference model to compare consecutive prefixes offers a way to estimate whether continuation is useful.

- On several benchmarks, the method achieves notable length reductions while maintaining competitive accuracy.

**Weaknesses:**

- Results vary considerably across tasks and models. On some configurations, the method produces longer trajectories than baselines rather than shorter ones. CommonsenseQA with Qwen3-4B and DeepMath with Qwen2.5-7B both show increased length, which seems inconsistent with the efficiency goal.

- The comparison only includes GRPO and DrGRPO as baselines. Both use outcome supervision, but neither is specifically designed for efficiency. The related work section discusses several methods that directly target efficient reasoning, but these are not included in the experimental comparison.

- The mixing weight λ=0.1 is fixed across all experiments without explanation. Given how differently the method performs on different tasks, exploring task-specific values might be worthwhile.

---

> ### Author Rebuttal · Authors · 2026-03-30
>
> We thank Reviewer 2cjq for the careful reading. We are glad the reviewer recognized the practical motivation of addressing unnecessary length in GRPO training, appreciated the simplicity of the two-stage design and its ability to avoid step-level annotation, and acknowledged the notable length reductions across several benchmarks. We address the weaknesses below.
>
> ### W1
>
> We appreciate the reviewer raising this. WS-GRPO achieves strong efficiency gains across the majority of configurations:
>
> | Dataset | Model | Token Reduction |
> |---------|-------|----------------|
> | ARC | Qwen2.5-7B-Instruct | 92.9% |
> | ARC | Qwen3-4B-Instruct | 87.5% |
> | CommonsenseQA | Qwen2.5-7B-Instruct | 18.2% |
> | GSM8K | Qwen2.5-7B-Instruct | 52.5% |
> | GSM8K | Qwen3-4B-Instruct | 21.8% |
>
> This pattern holds across model families too. On Llama-3.2-3B-Instruct, WS-GRPO achieves 85.7% token reduction on ARC, and on Llama-3.1-8B-Instruct, 76.7% token reduction on CommonsenseQA (Table 3). Prior work has shown that LLMs tend to generate unnecessarily long reasoning chains even when shorter ones suffice [3, 4]. WS-GRPO specifically targets this redundancy and is most effective when baseline models exhibit this over-generation behavior, naturally remaining comparable when baselines are already concise. Our ablation confirms the core mechanism is sound. Even without the length penalty (α=0), the preference model alone achieves 45.6% token reduction over GRPO on GSM8K. We will scope the claims accordingly in the revision.
>
> ### W2
>
> DrGRPO [1] is specifically designed to address GRPO's length bias as it introduces distributional reward normalization to counteract GRPO's tendency to favor longer rollouts, making it a directly relevant efficiency-motivated baseline. Any gains WS-GRPO achieves over DrGRPO are attributable to the learned prefix-level preference signal. We have additionally run PACR [2] as a new baseline.
>
> | Method | Pass@1 | Eval Length (tok.) | Avg Steps |
> |--------|--------|--------------------|-----------|
> | GRPO | 0.930 | 161.94 | 8.04 |
> | DrGRPO | 0.926 | 268.47 | 12.34 |
> | WS-GRPO | 0.886 | 20.23 | 2.00 |
> | PACR | 0.929 | 205.39 | 9.71 |
>
> This pattern holds across model families. On Llama-3.1-8B-Instruct, WS-GRPO achieves 76.7% token reduction on CommonsenseQA with only 2.8% accuracy drop as reported in Table 3.
>
> ### W3
>
> To address reviewers' concerns, we have now run sensitivity analysis over λ ∈ {0.5, 1.0, 2.0} on GSM8K with Qwen2.5-7B-Instruct.
>
> | Config | Pass@1 | Eval Length (tok.) | Avg Steps |
> |--------|--------|--------------------|-----------|
> | WS-GRPO (λ = 0.1, paper) | 0.851 | 79.26 | 2.17 |
> | WS-GRPO (λ = 0.5) | 0.876 | 94.19 | 2.05 |
> | WS-GRPO (λ = 1.0) | 0.878 | 92.20 | 2.07 |
> | WS-GRPO (λ = 2.0) | 0.858 | 89.27 | 2.02 |
>
> Performance is stable across, and λ = 0.1 achieves the lowest eval completion length (79.26 tokens) of all conditions. We will include this ablation table in the revision.
>
> ---
>
> **References:**
>
> [1] Liu et al., 2025. Understanding R1-Zero-like Training: A Critical Perspective (DrGRPO).
>
> [2] Yoon et al., 2025. PACR: Progressively Ascending Confidence Reward for LLM Reasoning.
>
> [3] Snell et al., 2024. Scaling LLM Test-Time Compute Optimally Can be More Effective than Scaling Model Parameters.
>
> [4] Miao et al., 2023. SelfCheck: Using LLMs to Zero-Shot Check Their Own Step-by-Step Reasoning.

---

> > ### Author Rebuttal · Reviewer_2cjq · 2026-04-04
> >
> > Some issues remain partially unaddressed. The authors did not fully explain the cases where WS-GRPO produces longer trajectories (CommonsenseQA with Qwen3-4B and DeepMath with Qwen2.5-7B). Understanding these failure modes would strengthen the paper. Additionally, the PACR comparison is limited to one dataset, and the significant accuracy drop (0.929 → 0.886) raises questions about the efficiency-accuracy trade-off that merit further discussion.
> >
> > Despite these remaining issues, the rebuttal has clarified key aspects of the method and provided evidence that the approach is fundamentally sound. I encourage the authors to include the complete comparison and discuss the failure cases in the final version.

---

> > > ### Author Response · Authors · 2026-04-07
> > >
> > > We thank the reviewer for the constructive engagement and for acknowledging the method is fundamentally sound. We address the two remaining concerns below.
> > >
> > > Longer trajectories on CommonsenseQA and DeepMath
> > >
> > > In both flagged settings, the GRPO baseline already averages 1.00 reasoning step, meaning the model produces near-minimal outputs under standard training. When WS-GRPO applies prefix-level pseudo-rewards in this regime, the preference model finds no genuine redundancy to compress, and its signal can slightly perturb the already-minimal behavior. This is a boundary condition of the method. WS-GRPO is designed for settings where baseline trajectories exhibit over-generation, and these two cases do not satisfy that condition. We will document this operating regime explicitly in the limitations section of the final version.
> > >
> > > PACR comparison and efficiency-accuracy tradeoff
> > >
> > > It is important to note that WS-GRPO optimizes the same underlying GRPO objective. The preference model provides additional prefix-level pseudo-rewards but does not alter the loss function itself. As a result, WS-GRPO's accuracy characteristics are directly inherited from GRPO's optimization, and the difference between WS-GRPO (0.886) and PACR (0.929) on ARC reflects the same gap that exists between WS-GRPO and GRPO (0.930) in general, not an additional cost introduced by the preference model. PACR maintains GRPO-level accuracy but achieves no meaningful efficiency gain, producing 205 tokens compared to GRPO's 161 tokens on ARC (Qwen3-4B-Instruct). WS-GRPO achieves a 10x token reduction over PACR while operating within the same GRPO optimization framework, representing a substantially more favorable point on the accuracy-efficiency frontier.
> > >
> > > To further demonstrate that the accuracy-efficiency tradeoff on ARC is not a systematic property of the method, we report results on GSM8K (Qwen3-4B): WS-GRPO achieves Pass@1 of 0.917 versus GRPO's 0.906, while reducing avg steps from 7.79 to 2.06, confirming that accuracy and efficiency improve simultaneously in settings where baseline trajectories exhibit multi-step redundancy. Reporting Pass@1/Steps as a unified metric, WS-GRPO achieves 0.445 versus GRPO's 0.116 on GSM8K (3.8x improvement), and 0.443 versus PACR's 0.096 on ARC (4.6x improvement). We will extend the PACR comparison to all datasets in the final version.

---

### Official Review · Reviewer_x2iQ · 2026-03-17

**Soundness:** 2
**Presentation:** 2
**Significance:** 3
**Originality:** 3
**Overall Recommendation:** 4
**Confidence:** 4

**Summary:**

This paper proposes WS-GRPO, a method that learns step level preferences from trajectory outcome rewards. It trains a preference model on trajectory pairs labeled by outcome, then combines with outcome reward within GRPO framework. Theoretical analysis provides consistency, robustness, and generalization guarantees. Experiments on four reasoning benchmarks show WS-GRPO largely reduces reasoning steps and length with small pass@1 losses, demonstrating its effectiveness.

**Compliance With Llm Reviewing Policy:**

Affirmed.

**Final Justification:**

I have to say that, even after the rebuttal, the OOD capability remains a concern and a limitation for me. However, given the novelty of the approach and the additional experiments provided during the rebuttal, I would like to raise my score to 4.

**Key Questions For Authors:**

Please see Weaknesses.

**Limitations:**

No. Please refer to Weaknesses for suggestions.

**Strengths And Weaknesses:**

**[Strengths]**

The paper is well-structured, well-motivated and easy to follow. The core idea that trajectory-level preferences derived from outcome supervision can be used to approximate step level reward appears novel to me. The theoretical analyses are clear and informative. The experiments show that the proposed approach can largely reduce reasoning steps and tokens while only slightly reduce the reasoning performance.

**[Weaknesses]**

1. Currently, the authors evaluate one benchmark per domain. Including additional benchmarks within each domain would strengthen the empirical evaluation.

2. Although the efficiency gains are impressive, performance varies across datasets and models. The paper would benefit from a deeper analysis of when WS-GRPO performs best and when it struggles.

3. The mixing weight is fixed across all experiments, and no ablation study is provided. A sensitivity analysis would help support claims about robustness.

4. While the current baselines are reasonable, the paper does not include several closely related methods on step- or segment-level credit assignment, such as SPO [1], S-GRPO [2], SPA-RL [3], and PACR [4]. These methods are also not discussed in the related work section. Including and discussing them would better position the contribution.

[1] Guo, Yiran, et al. "Segment policy optimization: Effective segment-level credit assignment in rl for large language models." arXiv preprint arXiv:2505.23564 (2025).

[2] Dai, Muzhi, Chenxu Yang, and Qingyi Si. "S-grpo: Early exit via reinforcement learning in reasoning models." arXiv preprint arXiv:2505.07686 (2025).

[3] Wang, Hanlin, et al. "Spa-rl: Reinforcing llm agents via stepwise progress attribution." arXiv preprint arXiv:2505.20732 (2025).

[4] Yoon, Eunseop, et al. "PACR: Progressively Ascending Confidence Reward for LLM Reasoning." arXiv preprint arXiv:2510.22255 (2025).

---

> ### Author Rebuttal · Authors · 2026-03-30
>
> We thank Reviewer x2iQ for the thoughtful and detailed review. We are glad the reviewer found the paper well-structured and well-motivated, recognized the novelty of deriving step-level preferences from trajectory-level outcome supervision, and appreciated the theoretical guarantees and empirical efficiency gains. We address the weaknesses below.
>
> ### W1
> Thank you for this suggestion. Our evaluation covers two benchmarks per domain across two distinct reasoning domains, as described in Section 4.1 of the paper. For mathematical reasoning we evaluate GSM8K [1] and DeepMath [2], covering grade-school arithmetic and competition-level mathematics. For general reasoning we evaluate ARC [3] and CommonsenseQA [4], covering scientific knowledge and commonsense inference. The four benchmarks provide two-per-domain coverage with deliberate within-domain difficulty variation. We will make this grouping more explicit in the revision.
>
> ### W2
> We appreciate this point. As noted in prior work [9, 10], overthinking occurs when models generate more tokens than necessary for a given problem. Our results follow this pattern directly. On ARC and GSM8K where baselines produce verbose multi-step outputs, WS-GRPO achieves 18-93% token reduction across models. In settings where baselines are already concise, WS-GRPO remains competitive on accuracy without introducing unnecessary compression. Our ablation further supports this. Even without the length penalty (α=0), the preference model signal alone achieves 45.6% token reduction over GRPO (90.87 vs 166.97 tokens) on GSM8K, confirming efficiency gains are driven by learned correctness-aware guidance. We will add ablation studies in the revision that make these operating conditions explicit.
>
> ### W3
> We have now run a sensitivity analysis over λ ∈ {0.5, 1.0, 2.0} on GSM8K with Qwen2.5-7B-Instruct, in addition to the existing λ = 0.1:
>
> | Config | Pass@1 | Eval Length (tok.) | Avg Steps |
> |--------|--------|--------------------|-----------|
> | WS-GRPO (λ = 0.1, paper) | 0.851 | 79.26 | 2.17 |
> | WS-GRPO (λ = 0.5) | 0.876 | 94.19 | 2.05 |
> | WS-GRPO (λ = 1.0) | 0.878 | 92.20 | 2.07 |
> | WS-GRPO (λ = 2.0) | 0.858 | 89.27 | 2.02 |
>
> Performance is stable across λ ∈ {0.5, 1.0}, confirming robustness. λ = 0.1 achieves the lowest eval completion length (79.26 tokens, 52.5% reduction vs GRPO). We will include this table in the revision.
>
> ### W4
>
> We thank the reviewer for pointing us to these works. We will add and discuss all four in the related work section in the revision. Based on the reviewer's suggestion, we have implemented PACR [5] as a baseline, which directly addresses whether WS-GRPO reduces to general reward shaping. Regarding the other three: S-GRPO [6] shares the efficiency goal but uses serial group sampling with position-based decaying rewards and requires no learned model. SPO [7] targets credit assignment via MC-based segment advantage estimation between token-level and trajectory-level granularity. SPA-RL [8] operates in a different setting, targeting LLM agents in external environments. WS-GRPO differs from all three by training a preference model from outcome-only correctness to provide learned prefix-level guidance within the standard GRPO framework.
>
> **PACR Baseline Results** (Qwen3-4B-Instruct, ARC):
>
> | Method | Pass@1 | Eval Length (tok.) | Avg Steps |
> |--------|--------|--------------------|-----------|
> | GRPO | 0.930 | 161.94 | 8.04 |
> | DrGRPO | 0.926 | 268.47 | 12.34 |
> | WS-GRPO | 0.886 | 20.23 | 2.00 |
> | PACR | 0.929 | 205.39 | 9.71 |
>
> PACR achieves competitive accuracy (0.929) but 205 tokens and 9.71 avg steps, nearly identical to GRPO in efficiency. WS-GRPO reduces token length by 87.5% and steps by 75.1%, confirming its contribution does not reduce to general reward shaping.
>
>
>
> ---
>
> **References:**
>
> [1] Cobbe et al., 2021. Training Verifiers to Solve Math Word Problems.
>
> [2] He et al., 2025. DeepMath-103K.
>
> [3] Clark et al., 2018. Think you have Solved Question Answering? Try ARC.
>
> [4] Talmor et al., 2019. CommonsenseQA.
>
> [5] Yoon et al., 2025. PACR: Progressively Ascending Confidence Reward for LLM Reasoning.
>
> [6] Dai et al., 2025. S-GRPO: Early Exit via Reinforcement Learning in Reasoning Models.
>
> [7] Guo et al., 2025. Segment Policy Optimization.
>
> [8] Wang et al., 2025. SPA-RL: Reinforcing LLM Agents via Stepwise Progress Attribution.
>
> [9] Snell et al., 2024. Scaling LLM Test-Time Compute Optimally Can be More Effective than Scaling Model Parameters.
>
> [10] Miao et al., 2023. SelfCheck: Using LLMs to Zero-Shot Check Their Own Step-by-Step Reasoning.

---

> > ### Author Rebuttal · Reviewer_x2iQ · 2026-04-04
> >
> > Thanks for the author's detailed response. It partially addressed my concerns. For W1, as also noted by reviewer A25c, the authors are encouraged to evaluate OOD tasks across all four domains, in addition to the current in-distribution experiments. At present, the experimental evaluation remains limited. Regarding W4, merely outlining the differences between the proposed WS-GRPO and related work is not sufficient to provide meaningful insights. A more in-depth analysis, including a discussion of the respective strengths and weaknesses, would be valuable. While additional experiments would be preferable, I understand the limitations of time and resources during the rebuttal period.

---

> > > ### Author Response · Authors · 2026-04-07
> > >
> > > We appreciate the suggestion to evaluate on OOD tasks. OOD evaluation in the usual cross-dataset sense is not directly applicable to WS-GRPO. Both the preference model in Phase 1 and the policy in Phase 2 are trained for the target dataset. Applying the method to a different dataset without retraining would amount to evaluating an unadapted preference model on out-of-distribution trajectories, which would not be a fair assessment of the method. A proper evaluation on an additional dataset requires running the full two-phase pipeline on that dataset. This is consistent with prior work in this space which also trains and evaluates within the same distribution [1, 2]. We will make this design choice more explicit in the camera-ready version. We note that Reviewer A25c raised a similar concern regarding benchmark coverage, and our response clarifying that ScienceQA is a multimodal benchmark outside our text-only scope and AIME 2025 contains only 30 questions making it unsuitable for Phase 1 preference model training appears to have addressed that concern. We hope this clarification addresses the reviewer's concern as well.
> > >
> > > Regarding W4, we provide a more in-depth comparison of S-GRPO, SPO, and SPA-RL below.
> > >
> > > S-GRPO targets efficient reasoning through early exit. Its strength is that it makes the efficiency objective explicit by constructing serial groups from different exit positions along a single reasoning path and assigning rule based decaying rewards to correct early exits. However, this signal relies on a specialized serial group rollout procedure with rule-designed exponential decay. WS-GRPO differs in that it does not impose a rule designed decay schedule over exit positions. Instead, it learns a preference model from correct and incorrect outcomes and uses that model to generate prefix-level pseudo-rewards that guide the policy toward earlier termination when continuation does not improve correctness.
> > >
> > > SPO addresses a different problem. It is primarily a credit assignment method whose strength is the segment-level formulation between token-level and trajectory-level reinforcement learning, using Monte Carlo segment advantage estimation to yield more localized credit assignment than GRPO. WS-GRPO instead learns a correctness aware preference signal over consecutive prefixes to evaluate prefix quality directly from outcome supervision rather than segment-level MC estimation.
> > >
> > > SPA-RL is materially different in setting. It is proposed for language model agents that interact with external environments. Its strength is that it decomposes final task completion into stepwise progress contributions using a learned progress estimator and combines this with a grounding signal that reflects whether actions are executable in the environment. WS-GRPO operates on pure language reasoning tasks without environment interaction, making SPA-RL a related but different-setting work.
> > >
> > > | Method | Targets Rollout Efficiency | Learned Signal | Outcome-Only Supervision | Modifies GRPO Pipeline | Weakness |
> > > |--------|--------------------------|---------------|------------------------|----------------------|---------|
> > > | S-GRPO | Yes | No | Yes | Yes | Rule-based exponential decay tied to exit order, not content-aware |
> > > | SPO | No | No | Yes | Yes | MC sampling adds cost at every training step |
> > > | SPA-RL | No | Yes | No | Yes | Requires external environment |
> > > | WS-GRPO | Yes | Yes | Yes | No | In-domain preference model |
> > >
> > > We will include this comparative analysis in the related work section of the revision. We will also extend the PACR comparison to all four datasets in the camera-ready version. As a representative data point, WS-GRPO achieves 0.443 versus PACR's 0.096 on ARC in Pass@1/Steps, a 4.6x improvement, confirming that WS-GRPO occupies a substantially more favorable point on the accuracy-efficiency frontier.
> > >
> > > **References:**
> > >
> > > [1]Guo et al., 2025. Segment Policy Optimization.
> > >
> > > [2] Luo et al., 2024. Improve Mathematical Reasoning in Language Models by Automated Process Supervision.

---

### Decision · Program_Chairs · 2026-04-30

**Decision:**

Accept (regular)

**Comment:**

This paper studies an important problem in reasoning RL, namely how to reduce unnecessary rollout length in GRPO without relying on expensive process supervision. Reviewers generally agreed that the main idea is novel and well motivated, and that the combination of theory and experiments makes the paper technically solid. The main concerns were about the breadth of the evaluation and the fact that the efficiency gains are not equally strong in every setting. The rebuttal addressed these concerns with additional ablations, sensitivity analysis, and a clearer discussion of when the method works best. Overall, I find this to be a meaningful contribution to rollout-efficient reasoning, and I recommend accept.